# MICE: Memory-driven Intrinsic Cost Estimation for Mitigating Constraint Violations

## Abstract

Constrained Reinforcement Learning (CRL) aims to maximize cumulative rewards while satisfying constraints. However, most existing CRL algorithms encounter significant constraint violations during training, limiting their applicability in safety-critical scenarios. In this paper, we identify the underestimation of the cost value function as a key factor contributing to these violations. To address this issue, we propose the Memory-driven Intrinsic Cost Estimation (MICE) method, which introduces intrinsic costs to enhance the cost estimate of unsafe behaviors, thus mitigating the underestimation bias. Our method draws inspiration from human cognitive processes, specifically the concept of flashbulb memory, where vivid memories of dangerous events are retained to prevent potential risks. MICE constructs a memory module to store unsafe trajectories explored by the agent. The intrinsic cost is formulated as the similarity between the current trajectory and the unsafe trajectories stored in memory, assessed by an intrinsic generator. We propose an extrinsic-intrinsic cost value function and optimization objective based on intrinsic cost, along with the corresponding optimization method. Theoretically, we provide convergence guarantees for the new cost value function and establish the worst-case constraint violation for the MICE update, ensuring fewer constraint violations compared to baselines. Extensive experiments validate the effectiveness of our approach, demonstrating a substantial reduction in constraint violations while maintaining policy performance comparable to baselines.

## 1 Introduction

Reinforcement learning (RL) has demonstrated great potential in numerous scenarios, such as video games Vinyals et al. (2019); Yu et al. (2022a), robotics control Haarnoja et al. (2018); Xu et al. (2020), and Go Schrittwieser et al. (2020), where agents explore the environment to learn the optimal policy that maximizes expected cumulative reward. However, the lack of safety considerations in RL leads to unsafe interactions with the environment, which are unacceptable in many safety-critical problems, such as robot navigation and autonomous driving. Constrained reinforcement learning (CRL) addresses this issue by finding optimal policies while satisfying predefined constraints, which extends the applicability of RL to real-world scenarios.

CRL is typically modeled as Constrained Markov Decision Processes (CMDP) Beutler & Ross (1985); Ross & Varadarajan (1989); Altman (2021), which integrates safety criteria in the form of constraints into RL, providing a fundamental mathematical framework. Current CRL methods can be broadly categorized into primal and primal-dual methods. Primal-dual methods Tessler et al. (2018); Yu et al. (2019); Paternain et al. (2019); Ding et al. (2020) convert the constrained problem into an unconstrained one using the Lagrangian function and solve it in the dual space. However, these methods often suffer from inherent oscillations. PID Lagrangian Stooke et al. (2020) introduces proportional and differential control to mitigate these oscillations. Primal methods, such as Constrained Policy Optimization (CPO) Achiam et al. (2017), approximate the constrained optimization problem with surrogate functions in primal space. Despite these advancements, state-of-the-art CRL methods still experience significant constraint violations during training.

In this work, we aim to mitigate constraint violations during training in CRL by addressing a critical issue: the underestimation of the cost value function. We identify the underestimation as a key factor contributing to constraint violations. Overestimation bias is common in value functions of RL

due to the maximization of noisy value estimates Thrun & Schwartz (2014); Fujimoto et al. (2018), and this noise is unavoidable in function approximation methods. Additionally, temporal difference methods accumulate errors by updating value function estimates using subsequent state estimates. In CRL, the cost value function requires minimization when constraints are violated, in contrast to the maximization operation in RL value function updates. This difference introduces unique challenges: underestimating costs can make risky actions appear less costly, leading to frequent constraint violations, even with the optimization methods capable of finding the optimal policy.

Drawing inspiration from human cognitive processes, we propose the Memory-driven Intrinsic Cost Estimation (MICE) algorithm, which constructs an extrinsic-intrinsic cost value update to enhance the cost estimate of unsafe trajectories, thus mitigating underestimation. Psychological studies of flashbulb memory Conway (2013) reveal that humans vividly remember significant and surprising events, helping them develop cautionary behaviors, such as avoiding fire after a burn. Similarly, we equip CRL agents with a flashbulb memory module to enhance their risk awareness, storing unsafe trajectories explored by the agent. We introduce an intrinsic cost generated from the flashbulb memory, which is formulated as the similarity between the current trajectory and the unsafe ones stored in memory. Here, extrinsic costs denote task-related costs in CRL, allowing for clear differentiation. We propose an extrinsic-intrinsic update formulation of the cost value function, effectively mitigating the underestimation by enhancing the cost estimate of unsafe behaviors. Based on the extrinsic-intrinsic cost value function, we propose an optimization objective within the trust region and provide the corresponding optimization method. Theoretically, we establish a constraint bound for the extrinsic-intrinsic cost value function and provide a worst-case constraint violation for the MICE update, ensuring few constraint violations during training. Additionally, we provide a convergence analysis for the extrinsic-intrinsic cost value function. Comparison experiments with baselines demonstrate that our method significantly reduces constraint violations while maintaining robust policy performance. Our contribution can be summarized as follows:

- We identify that the underestimation of the cost value function is prevalent in CRL, which is a key factor in constraint violations during training. This insight highlights an important challenge for ensuring safety and reliability in CRL applications.

- We propose the MICE algorithm to mitigate underestimation, incorporating the extrinsic-intrinsic cost value estimate and corresponding optimization objective.

- We provide theoretical guarantees for MICE, including constraint bounds and convergence analysis. Extensive experiments demonstrate that MICE significantly reduces constraint violations while maintaining policy performance compared to baselines.

## 2 RELATED WORK

**CRL methods.** The optimization methods in CRL can be classified into two categories: the primal-dual method and the primal method. Primal-dual methods Ding et al. (2021); Ying et al. (2024) convert constrained problems into unconstrained ones via introducing dual variables. Tessler et al. (2018) introduce multi-timescale Lagrangian methods to guide the policy update toward constraint satisfaction. Theoretically, Ding et al. (2020) establish the global convergence with sublinear rates regarding the optimality gap. Stooke et al. (2020) introduce proportional and differential control to mitigate cost overshoot and oscillations in the learning dynamics. However, primal-dual approaches remain sensitive to initial parameters, limiting their application Zhang et al. (2022). In contrast, primal methods directly optimize constrained problems in the primal space Chow et al. (2018); Yu et al. (2022b). Chow et al. (2019) propose a safe linear programming algorithm based on a Lyapunov approach to solve the constrained problems. CPO Achiam et al. (2017) provides a lower bound on performance and an upper bound on constraint violation. PCPO Yang et al. (2020) first improves the reward within the trust region, then projects the policy to the feasible region. FOCOPS Zhang et al. (2020) solves the constraint problem in the nonparametric policy space and then projects the updated policy back into the parametric space. CUP Yang et al. (2022) provides generalized theoretical guarantees for surrogate functions with generalized advantage estimator Schulman et al. (2015), effectively reducing variance while maintaining acceptable bias. Due to the underestimation of the cost value function, all of the above methods are unable to avoid significant constraint violations during the training process. We design a safety-based intrinsic cost to mitigate the underestimation, thus achieving few constraint violations while maintaining a similar performance as baselines.

**Overestimation in RL.** The issue of overestimation in RL has been extensively studied. Double Q-learning Hasselt (2010) addresses this problem by employing two independent estimators to decouple action selection and evaluation. Double DQN Van Hasselt et al. (2016) extends this concept to function approximation, utilizing a separate target value function to estimate the value of the current policy, thus reducing bias by enabling evaluating actions without maximization bias. However, in actor-critic frameworks, the slow-changing nature of the policy means that the current and target value estimates often remain close, failing to eliminate maximization bias. To address this, TD3 Fujimoto et al. (2018) selects the minimum value from a pair of critics, thereby reducing overestimation. AdaEQ Wang et al. (2021) employs the ensemble method, adjusting the ensemble size based on Q-value approximation error to mitigate overestimation. In this paper, we demonstrate that the underestimation of cost value in CRL causes constraint violations during training, and we introduce memory-driven intrinsic cost to effectively mitigate underestimation.

**Intrinsic reward.** Intrinsic rewards are typically used to design exploration strategies in RL, generally falling into two categories. The first category encourages agents to explore novel states Zhang et al. (2021); Seo et al. (2021). The second category incentivizes behaviors aimed at reducing prediction errors or uncertainties to improve the agent's understanding of the environment Sharma et al. (2019); Laskin et al. (2022). Lipton et al. (2016) indicate that agents tend to periodically revisit states under new policies after forgetting them, introducing an intrinsic fear model to prevent periodic catastrophes. In CRL tasks, ROSARL Tasse et al. (2023) treats constraints as intrinsic rewards, optimizing policies by determining the minimal penalty for unsafe states. In this paper, we use intrinsic costs for safer exploration when extrinsic costs are underestimated. Additionally, intrinsic costs generated from memory provide anticipatory signals for policy updates to avoid dangerous regions that have been explored.

## 3 PRELIMINARY

RL can be modeled as a Markov decision process (MDP), denoted by a tuple $(S, A, R, P, \rho, \gamma)$, where $S$ is the state space, $A$ is the action space, $R : S \times A \to \mathbb{R}$ is the reward function, $P : S \times A \to [0, 1]$ is the transition probability function, $\rho$ is the initial state distribution, and $\gamma \in (0, 1)$ is the discount factor of the reward. Starting from an initial state $s_0$ sampled from the initial state distribution $\rho$, the agent perceives the state $s_t$ from the environment at each time step $t$, and takes the action $a_t$ sampled from the policy $\pi : S \to A$, receives the reward $r_t = R(s_t, a_t)$, and transfers to the next state $s_{t+1}$ according to $P(s_{t+1}|s_t, a_t)$. $\Pi$ is the set of all stationary policies. The discounted future state visitation distribution is defined as $d^\pi(s) := (1 - \gamma) \sum_{t=0}^{\infty} \gamma^t P(s_t = s|\pi)$. The agent aims to find the optimal policy by maximizing the expected discounted return $J_R(\pi) := \mathbb{E}_{\tau \sim \pi}[\sum_{t=0}^{\infty} \gamma^t R(s_t, a_t)]$, where $\tau = (s_0, a_0, s_1, a_1, \cdots)$ is the trajectory based on the policy $\pi$. The value function based on policy $\pi$ is $V_R^\pi(s) := \mathbb{E}_{\tau \sim \pi}[\sum_{t=0}^{\infty} \gamma^t R(s_t, a_t)|s_0 = s]$, and action-value function is $Q_R^\pi(s, a) := \mathbb{E}_{\tau \sim \pi}[\sum_{t=0}^{\infty} \gamma^t R(s_t, a_t)|s_0 = s, a_0 = a]$. The advantage function measures the advantage of action $a$ over the mean value: $A_R^\pi(s, a) := Q_R^\pi(s, a) - V_R^\pi(s)$.

A Constrained Markov Decision Process (CMDP) $(S, A, R, C, P, \rho, \gamma)$ introduces constraints to the MDP to restrict the set of allowable policies. $C : S \times A \to \mathbb{R}$ denotes the extrinsic cost function, which maps the state-action pairs to extrinsic costs. We distinguish the intrinsic and extrinsic costs by $c^I$ and $c^E$, respectively. The extrinsic costs refer to constraints in the actual task. $d$ denotes the constraints threshold, and the expected cumulative discount cost is desired to satisfy $J_C(\pi) := \mathbb{E}_{\tau \sim \pi}[\sum_{t=0}^{\infty} \gamma^t C(s_t, a_t)] \le d$. The cost value function $V_C^\pi(s)$, cost action value function $Q_C^\pi(s, a)$ and cost advantage function $A_C^\pi(s, a)$ in CMDP can be obtained as in MDP by replacing the reward $R$ with the cost $C$. The CRL aims to find an optimal policy by maximizing the expected discount return over the set of feasible policies $\Pi_C := \{\pi \in \Pi : J_C(\pi) \le d\}$:

$$\arg \max_{\pi \in \Pi} J_R(\pi)$$
$$s.t. \quad J_C(\pi) \le d \tag{1}$$

## 4 METHODOLOGY

In this section, we introduce the Memory-driven Intrinsic Cost Estimation (MICE) algorithm. We first present the underestimation of the cost value function in CRL. Then we construct the flashbulb memory to store unsafe trajectories and the intrinsic cost generator to correct the underestimation.

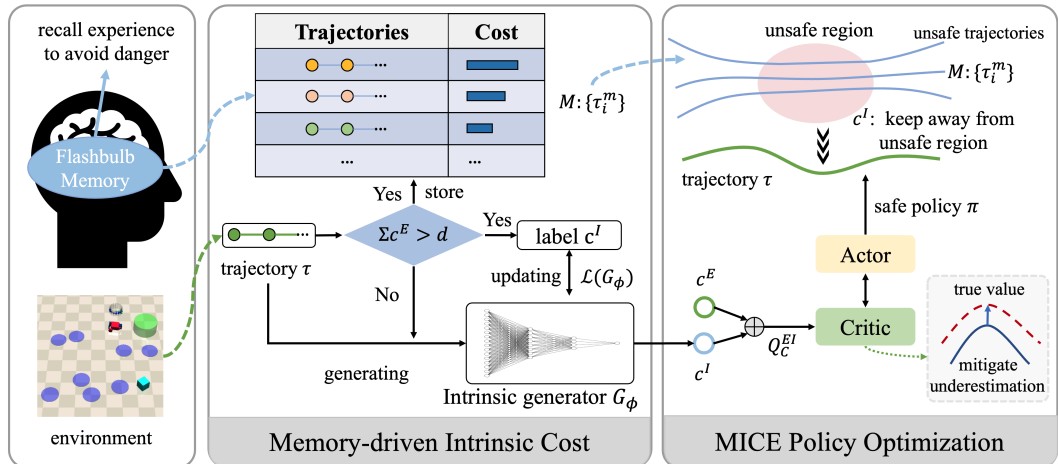

Figure 1: Structure of MICE.

Finally, we propose an extrinsic-intrinsic update formulation for the cost value in MICE and a new optimization objective with the solution.

### 4.1 UNDERESTIMATE BIAS IN COST VALUE FUNCTION

Overestimation commonly arises in RL because updates to the value function tend to greedily select high action values, resulting in estimations that exceed the optimal value Fujimoto et al. (2018). Conversely, in the estimation of cost values for CRL, especially when constraints are violated, there is a tendency to minimize costs, which results in an underestimation of the cost value function.

In cost value estimation methods such as Q-learning, a greedy strategy is used to update the cost value function $Q_C(s,a) \leftarrow Q_C(s,a) + \alpha[c + \gamma \min_{a'} Q_C(s',a') - Q_C(s,a)]$ during constraint violations. Assuming the value estimation contains zero-mean noise $\epsilon$, a consistent underestimation bias is induced by minimizing the noisy value estimate $Q_C(s',a') + \epsilon$. The zero-mean property of noise is disrupted after minimization, then the minimization of the value estimate is generally smaller than the true minimization $\mathbb{E}_\epsilon[\min_{a'} Q_C(s',a') + \epsilon] \leq \min_{a'} Q_C(s',a')$ Thrun & Schwartz (2014). Noise errors in function approximation methods are unavoidable Fujimoto et al. (2018).

In CRL methods based on actor-critic architecture, the policy learns from value estimations produced by the approximate critic and cost critic. When constraints are violated, the policy is updated with a policy gradient in the direction that minimizes the expectation of cost value estimate: $\arg\min_{\pi \in \Pi_\theta} \mathbb{E}_{s \sim d^\pi, a \sim \pi}[Q_C(s,a)]$, where $\Pi_\theta$ represents the policy set parameterized by $\theta$. Denote the true cost value function as $Q_C(s,a)$ and the approximate cost value function as $\hat{Q}_C(s,a)$. Updated from the current policy $\pi_k(\cdot|\theta)$ with the deterministic policy gradient, we denote the policy derived from the true cost value $Q_C(s,a)$ as $\pi$, and the policy derived from the approximate cost value $\hat{Q}_C(s,a)$ as $\hat{\pi}$. According to TD3 Fujimoto et al. (2018), if the approximation is lower than the true value due to unavoidable noise in the function approximation: $\mathbb{E}[\hat{Q}_C(s,\pi(s))] \leq \mathbb{E}[Q_C(s,\pi(s))]$, then the cost value is underestimated under the updated policy $\hat{\pi}$ within a sufficiently small step size:

$$\mathbb{E}[\hat{Q}_C(s,\hat{\pi}(s))] \leq \mathbb{E}[Q_C(s,\hat{\pi}(s))] \tag{2}$$

To validate the issue of underestimation bias, we compare cost value estimates for various states against their corresponding true values in CPO Achiam et al. (2017) and PID Lagrangian Stooke et al. (2020). The true value is estimated using the average discount constraint over 1,000 episodes under the current policy. Experimental results in Figure 2 show that the cost value functions of different CRL methods are significantly underestimated across various environments during the learning process. The underestimation bias can be propagated and accumulated through temporal difference updates, as the underestimated cost value estimate serves as the target for subsequent updates.

Compared to overestimation in RL, underestimation in CRL has more detrimental impacts, which generates unsafe actions that cause damage or task failure. In CRL, actions yielding high rewards

but violating constraints are mistakenly perceived safe by an underestimated critic and are subsequently selected. These unsafe actions propagate through the Bellman equation, generating even more unsafe actions as the underestimation bias increases. This explains the constraint violations during training in various CRL methods.

## 4.2 Intrinsic Cost Generated from Flashbulb Memory

Risk awareness helps humans identify potential dangers and adopt conservative behaviors to ensure safety. Typically, humans are impressed by previous risky behaviors or experiences, which are vividly recalled to circumvent danger in similar scenarios. However, CRL agents with underestimated critics often fail to recognize the consequences of unsafe actions, leading to constraint violations. Inspired by human cognitive mechanisms, we introduce an intrinsic generator that outputs memory-driven intrinsic cost signals to enhance the agent's risk awareness.

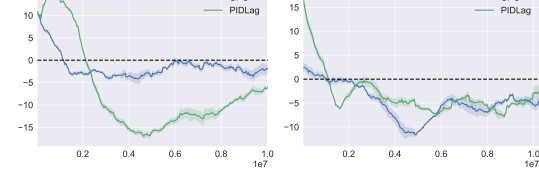

Figure 2: Underestimation error in different environments. The x-axis denotes the time step, the y-axis is the cost value estimate minus the true value, and the dashed line is the zero deviation.

We construct a flashbulb memory to store unsafe trajectories where the cumulative extrinsic cost exceeds the constraint threshold. These trajectories are organized as Markov chains with state-action pairs and cumulative costs. The memory capacity is fixed, mirroring the human tendency to prioritize the most dangerous and recent experiences. When the memory capacity is reached, the earliest trajectories are sorted by cumulative costs, and the one with the smallest value is removed to store a new unsafe trajectory. This mechanism ensures the memory remains relevant to the current policy while retaining the most significant experiences.

We propose an intrinsic cost $c^I$ derived from the flashbulb memory. Denote the flashbulb memory as $M : \{\tau_0^m, \cdots, \tau_i^m, \cdots, \tau_{n-1}^m\}$ with $n$ unsafe trajectories, where $\tau_i^m : \{s_0, a_0, \cdots\}$ represents the $i$-th unsafe trajectory stored in memory. Denote the current trajectory rollout from the initial to the time step $t$ as $\tau(t) : \{s_0, a_0, \cdots, s_{t-1}, a_{t-1}\}$. Intrinsic costs $c_t^I$ at time step $t$ is generated by comparing the difference between the current trajectory and the unsafe trajectories in memory $M$:

$$c_t^I = \frac{h\gamma_I^k}{(1 + e^{l_2(t)})}, \quad where \quad l_2(t) = \sum_{i=0}^{n-1} \|W(\tau_i^m(t) - \tau(t))\|_2 \qquad (3)$$

where $t$ denotes the time steps of agent's rollout in environment, $\tau_i^m(t) = \tau_i^m[0 : t - 1]$ denotes the segment of the $i$-th unsafe trajectory from the initial to the time step $t$. $\gamma_I \sim (0, 1)$ denotes the intrinsic discount factor, operating with the iteration number $k$ of the value function update. $h$ is the intrinsic factor. The weight vector $W = [w_0, w_1, \cdots]$ assigns weights to different state-action pairs, where pairs with a positive extrinsic cost $c^E$ are given greater weight to account for their significant influence on the cumulative costs:

$$w_t = \begin{cases} 1, & \text{if} \quad c_t^E > 0 \\ \omega, & \text{if} \quad c_t^E = 0, \quad 0 \le \omega \le 1 \end{cases} \qquad (4)$$

The Euclidean distance $l_2$ is computed between the current trajectory and unsafe trajectories in memory $M$, with the accumulation serving as a measure of divergence. A smaller divergence indicates that the current trajectory is more similar to previous unsafe experiences, resulting in a higher risk of constraint violation and a greater intrinsic cost $c^I$.

To minimize memory accesses, we design an intrinsic generator $G_\phi(\tau)$, parameterized by a network, to generate intrinsic costs $c^I$ based on flashbulb memory. The generator $G_\phi(\tau)$ takes the current trajectory as input and outputs an intrinsic cost signal. A random projection layer within the generator compresses the trajectories into latent space, reducing data dimensionality while capturing relations between state-action pairs Zhu et al. (2020). During the agent's learning process, if the cumulative extrinsic cost of the current trajectory $\tau$ is below the constraint threshold $d$, generator $G_\phi$ directly generates the intrinsic cost. Otherwise, the trajectory $\tau$ is stored in memory $M$, and $G_\phi$ is updated

according to following loss function, which regresses towards the corresponding intrinsic cost labels $c^I$, as defined in Equation 3:

$$\mathcal{L}(G_\phi) = \mathbb{E}_\tau (G_\phi(\tau) - c^I)^2, \quad if \quad \sum c^E > d \tag{5}$$

## 4.3 Safety Policy Optimization with the Intrinsic Cost

The cost value function $Q_C(s,a)$ updated with only extrinsic costs $c^E$ in CRL of Q-learning is: $Q_C(s,a) = (1-\alpha)Q_C(s,a) + \alpha(c^E + \gamma \min_{a'} \mathbb{E}_{s'}[Q_C(s',a')])$. To mitigate the underestimation, we propose a new update of the extrinsic-intrinsic cost value function $Q_C^{EI}(s,a)$, which incorporates both memory-driven intrinsic costs and task-driven extrinsic costs:

$$Q_C^{EI}(s,a) = (1-\alpha)Q_C^{EI}(s,a) + \alpha(c^E + c^I + \gamma \min_{a'} \mathbb{E}_{s'}[Q_C^{EI}(s',a')]) \tag{6}$$

where $c^E$ and $c^I$ denote extrinsic and intrinsic costs, respectively. Starting from the same initialization value, $Q_C^{EI}$ is greater than the $Q_C$ under the same state-action pair: $Q_C^{EI}(s,a) \geq Q_C(s,a)$, since $Q_C^{EI}$ has a larger update target.

By augmenting the agent's memory, the extrinsic-intrinsic target cost value increases the cost estimate of the state-action pair, effectively mitigating the underestimation. The extrinsic-intrinsic value function potentially introduces overestimation. It is important to note that overestimation does not result in constraint violations compared to underestimation in CRL. Additionally, the propagation of overestimation through cost value updates is limited, as the policy tends to avoid actions with high-cost estimates Fujimoto et al. (2018). Moreover, overestimation within our extrinsic-intrinsic value function can effectively correct the estimation bias in high-value regions Karimpanal et al. (2023), see Appendix B.1.

Based on the extrinsic-intrinsic cost value function, we define the cumulative discount extrinsic-intrinsic cost as $J_C^{EI}(\pi) := \mathbb{E}_{\tau \sim \pi}[\sum_{t=0}^{\infty} \gamma^t C^{EI}(s_t,a_t)] = \mathbb{E}_{\tau \sim \pi}[\sum_{t=0}^{\infty} \gamma^t (c^E + c^I)]$, where $C^{EI}(s,a) = c^E + c^I$ is the extrinsic-intrinsic cost function. The extrinsic-intrinsic advantage function in MICE is defined as: $A_C^{EI}(s,a) = \mathbb{E}_{s'}[c^E + c^I + \gamma V_C(s') - V_C(s)]$. To reduce constraint violations, we replace $J_C$ with the extrinsic-intrinsic constraint $J_C^{EI}$ in the optimization objective. To facilitate optimization, we give the difference in expectation constraint of extrinsic-intrinsic $J_C^{EI}(\pi')$ and extrinsic $J_C(\pi)$.

**Lemma 1.** *Given arbitrary two policies $\pi$ and $\pi'$, the difference in expectation constraint of extrinsic-intrinsic $J_C^{EI}(\pi')$ and extrinsic $J_C(\pi)$ can be expressed as:*

$$J_C^{EI}(\pi') - J_C(\pi) = \mathbb{E}_{\tau|\pi'}\left[\sum_{t=0}^{\infty} \gamma^t A_C^{EI}(s_t,a_t|\pi)\right] \tag{7}$$

*where $A_C^{EI}(s_t,a_t|\pi) = \mathbb{E}_{s_{t+1}}[c_t^E + c_t^I + \gamma V_C^\pi(s_{t+1}) - V_C^\pi(s_t)]$. The expectation is taken over trajectories $\tau$, and $\mathbb{E}_{\tau|\pi'}$ indicates that actions are sampled from $\pi'$ to generate $\tau$.*

A proof is provided in Appendix B.2. According to equation 7, we give the optimization objective of MICE:

$$\pi_{k+1} = \arg\max_{\pi \in \Pi_\theta} \mathbb{E}_{s \sim d^\pi, a \sim \pi}[A_R^{\pi_k}(s,a)]$$

$$s.t. \quad J_C(\pi_k) + \frac{1}{1-\gamma}\mathbb{E}_{s \sim d^\pi, a \sim \pi}[A_C^{EI}(s,a|\pi_k)] \leq d \tag{8}$$

However, the complex dependency of state visitation distribution $d^\pi(s)$ on unknown policy $\pi$ makes equation 8 difficult to optimize directly. This paper uses the samples generated by the current policy $\pi_k$ to approximate the original problem in the trust region. Based on the extrinsic-intrinsic value update function, we seek to solve the following optimization problem:

$$\pi_{k+1} = \arg\max_{\pi \in \Pi_\theta} \mathbb{E}_{s \sim d^{\pi_k}, a \sim \pi}[A_R^{\pi_k}(s,a)]$$

$$s.t. \quad J_C(\pi_k) + \frac{1}{1-\gamma}\mathbb{E}_{s \sim d^{\pi_k}, a \sim \pi}[A_C^{EI}(s,a|\pi_k)] \leq d \tag{9}$$

$$D(\pi\|\pi_k) \leq \delta$$

where $D(\pi\|\pi_k) = \mathbb{E}_{s\sim d^{\pi_k}}[D_{KL(\pi\|\pi_k)}[s]]$, $D_{KL}$ is the KL divergence and $\delta > 0$ is the step size. The set $\{\pi \in \Pi_\theta : D(\pi\|\pi_k) \leq \delta\}$ is the trust region.

The MICE-CPO method is proposed to solve the optimization objective 9. We approximate the reward objective and cost constraints with first-order expansion and approximate the KL-divergence constraint with second-order expansion. The local approximation to equation 9 is:

$$\theta_{k+1} = \arg\max_\theta g^T(\theta - \theta_k)$$
$$s.t. \quad c + (g_C^{EI})^T(\theta - \theta_k) \leq 0 \tag{10}$$
$$\frac{1}{2}(\theta - \theta_k)^T H(\theta - \theta_k) \leq \delta$$

where $g$ is the gradient of the reward objective and $g_C^{EI}$ is the gradient of extrinsic-intrinsic constraint in 9, $c = J_C(\pi_k) - d$, $H$ is the Hessian of the KL-divergence. When the constraint is satisfied, we can get the analytical solution with the primal-dual method. The solution to the primal problem is:

$$\theta^* = \theta_k + \frac{1}{\lambda^*}H^{-1}(g - g_C^{EI}\nu^*) \tag{11}$$

where $\lambda$ and $\nu$ are the Lagrangian multipliers of the KL-divergence term and the constraint term in the Lagrangian function, respectively. $\lambda^*$, $\nu^*$ are the solutions to the dual problem:

$$\nu^* = \max\{0, \frac{\lambda^* c - u}{v}\}, \quad \lambda^* = \arg\max_{\lambda \geq 0}
\begin{cases}
\frac{1}{2\lambda}\left(\frac{u^2}{v} - q\right) + \frac{\lambda}{2}\left(\frac{c^2}{v} - \delta\right) - \frac{uc}{v}, & \text{if}\lambda c > u \\
-\frac{1}{2}\left(\frac{q}{\lambda} + \lambda\delta\right), & \text{otherwise,}
\end{cases} \tag{12}$$

where $q = g^T H^{-1} g$, $u = g^T H^{-1} g_C^{EI}$, $v = (g_C^{EI})^T H^{-1} g_C^{EI}$. When the constraint is violated, we use the conjugate gradient method Achiam et al. (2017) to decrease the constraint value:

$$\theta^* = \theta_k - \left(\frac{2\delta}{(g_C^{EI})^T H^{-1} g_C^{EI}}\right)^{\frac{1}{2}} H^{-1} g_C^{EI} \tag{13}$$

We also provide the MICE-PIDLag optimization method, detailed in Appendix A.3.

## 4.4 THEORETICAL ANALYSIS

For extrinsic-intrinsic constraints in MICE, we give an upper bound on the constraint difference:

**Theorem 1** (Extrinsic-intrinsic Constraint Bounds). *For arbitrary two policies $\pi'$ and $\pi$, the following bound for cumulative discount extrinsic-intrinsic cost holds:*

$$J_C^{EI}(\pi') - J_C^{EI}(\pi) \leq \frac{1}{1-\gamma}\mathbb{E}_{s\sim d^\pi, a\sim\pi'}\left[A_C^{EI}(s, a|\pi) + \frac{2\gamma\epsilon_{\pi'}^{EI}}{1-\gamma}D_{TV}(\pi'\|\pi)[s]\right] \tag{14}$$

*where $\epsilon_{\pi'}^{EI} := \max_s |\mathbb{E}_{a\sim\pi', s'\sim P}[\varepsilon_V^{EI}(s, a, s')]|$, $\varepsilon_V^{EI}(s, a, s') = C^{EI}(s, a, s') + \gamma V_C(s') - V_C(s)$ denotes the extrinsic-intrinsic TD-error, $D_{TV}(\pi'\|\pi)[s] = (1/2)\sum_a |\pi'(a|s) - \pi(a|s)|$.*

The proof is provided in Appendix B.3. The upper bound in Theorem 1 is associated with the TV divergence between $\pi$ and $\pi'$. A larger divergence between these two policies results in a larger upper bound on the constraint gap. This theorem explains the optimization objective 9 within the trust region in MICE.

By mitigating the underestimation, the MICE algorithm significantly reduces constraint violations during the learning process. We further establish a theoretical upper bound on the constraint violation for the updated policy within the optimization framework of MICE:

**Theorem 2** (MICE Update Worst-Case Constraint Violation). *Suppose $\pi_k$, $\pi_{k+1}$ are related by the optimization objective 9, an upper bound on the constraint of the updated policy $\pi_{k+1}$ is:*

$$J_C(\pi_{k+1}) \leq d - I + \frac{\sqrt{2\delta}\gamma\epsilon_C^{\pi_{k+1}}}{(1-\gamma)^2} \tag{15}$$

*where $\epsilon_C^{\pi_{k+1}} := \max_s |\mathbb{E}_{a\sim\pi_{k+1}}[A_C^{\pi_k}(s, a)]|$, $I = \mathbb{E}_{\tau|\pi_{k+1}}\left[\sum_{t=0}^\infty \gamma^t c_t^I\right]$.*

A proof is provided in Appendix B.3. We further analyze this upper bound in Appendix C.2.1, which is related to the intrinsic factor. Theorem 2 demonstrates that our method achieves a tighter upper bound on constraint violation compared to CPO, guaranteeing that the updated policy in MICE has a lower probability of exceeding the constraint threshold. Based on similar assumptions as in TD3 and Double Q-learning, we give convergence guarantees of the extrinsic-intrinsic cost value function.

**Theorem 3** (Convergence Analysis). *Given the following conditions: (1) Each state-action pair is sampled an infinite number of times. (2)The MDP is finite. (3) $\gamma \in [0, 1)$. (4) $Q_C^{EI}$ values are stored in a lookup table. (5) $Q_C^{EI}$ receives an infinite number of updates. (6) The learning rates satisfy $\alpha_t(s, a) \in [0, 1]$, $\sum_t \alpha_t(s, a) = \infty$, $\sum_t (\alpha_t(s, a))^2 < \infty$ with probability 1 and $\alpha_t(s, a) = 0$, $\forall(s, a) \neq (s_t, a_t)$. (7) $Var[c_t^E + c_t^I] < \infty, \forall s, a$. The extrinsic-intrinsic $Q_C^{EI}$ will converge to the optimal value function $Q_C^*$ with probability 1.*

The proof is in Appendix B.4. Theorem 3 ensures that our method converges to the optimal solution.

## 5 EXPERIMENT

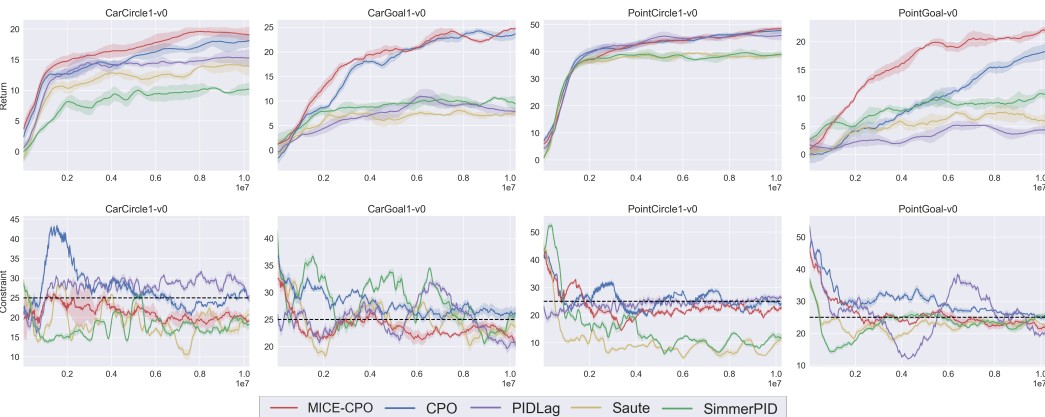

Figure 3: Comparison of MICE to baselines on Safety Gym. The x-axis is the total number of training steps, the y-axis is the average return or constraint. The solid line is the mean and the shaded area is the standard deviation. The dashed line is the constraint threshold which is 25.

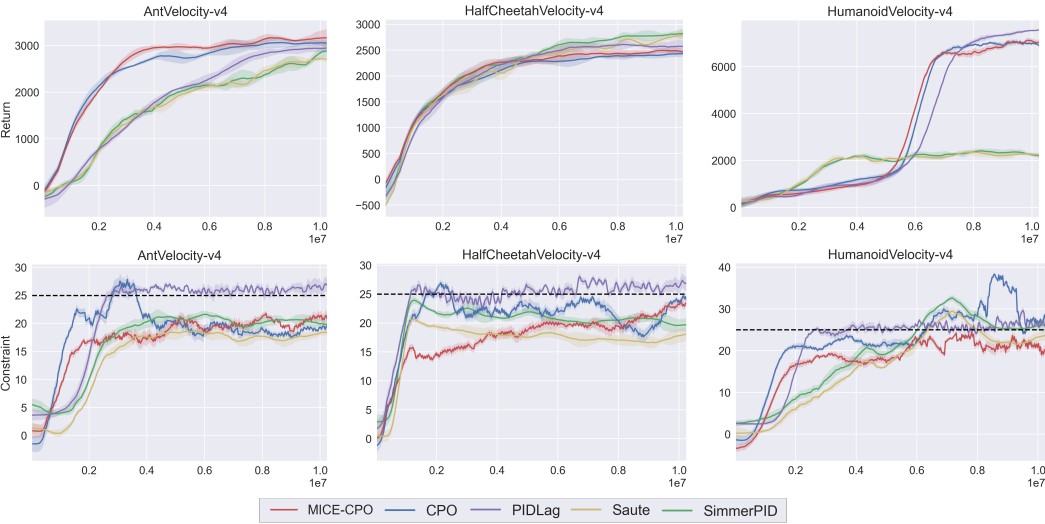

Figure 4: Comparison of MICE to baselines on Safety MuJoCo. The x-axis is the total number of training steps, the y-axis is the average return or constraint.

The experiments aim to answer the following questions: 1) Does MICE reduce constraint violations during training while maintaining policy performance compared to baselines? 2) Does the intrin-

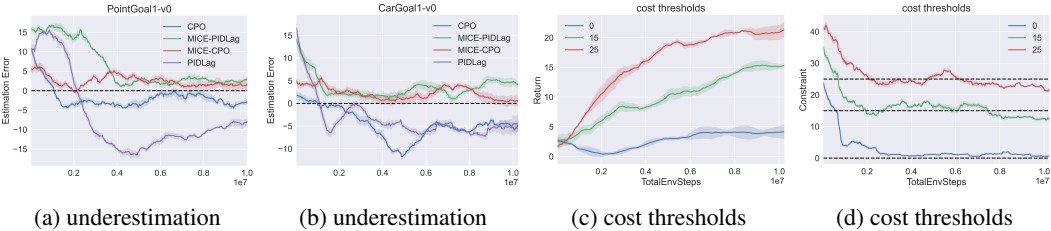

(a) underestimation     (b) underestimation     (c) cost thresholds     (d) cost thresholds

Figure 5: **(a)(b)** Validation experiments of mitigating underestimation with MICE. The y-axis is the cost value estimate minus the true value, and the dashed line is the zero deviation. **(c)(d)** Robustness of MICE to different cost thresholds.

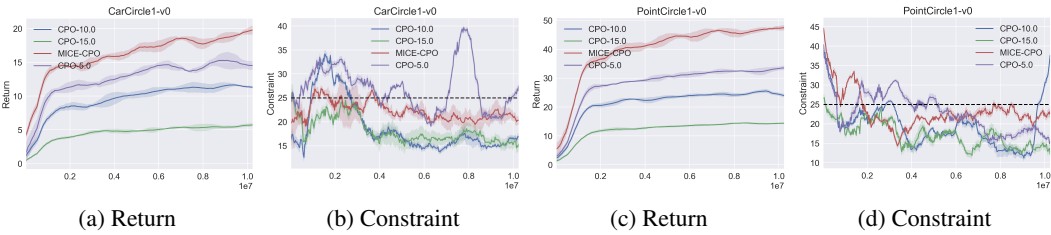

(a) Return     (b) Constraint     (c) Return     (d) Constraint

Figure 6: Ablation experiments of extrinsic-intrinsic cost value in MICE. Comparing the MICE algorithm with versions that directly add various constants (5, 10, 15) into the cost value function.

sic cost component effectively mitigate underestimation? We conducted experiments across four navigation tasks using the OpenAI Safety Gym Brockman et al. (2016) and three MuJoCo physical simulator tasks Todorov et al. (2012). Baselines include the primal-dual method PID Lagrangian Stooke et al. (2020), the primal method CPO Achiam et al. (2017), and state augmentation methods Saute Sootla et al. (2022a) and Simmer Sootla et al. (2022b), which focus on zero constraint violations. Experimental results for more baselines are provided in the Appendix C.2.3. We implemented the MICE-CPO and MICE-PIDLag methods, with detailed optimization procedures outlined in Appendix A. All experiments were conducted under uniform conditions to ensure fairness and reproducibility. The total training time step is $10^7$, with a maximum trajectory length of 1000 steps. To reduce randomness, we used 6 random seeds for each method, calculating the mean and variance of the results. Additional experiments are provided in Appendix C.2, and our code can be found in Appendix C.1.

**Environments Description.** All tasks aim to maximize the expected reward (the higher, the better) while satisfying the constraint (the lower, the better). In Safety Gym, we train Point and Car agents on navigation tasks, including the Goal task to navigate to a goal while avoiding hazards, and the Circle task to go around the center of the circle area without crossing boundaries. In Safety MuJoCo, agents receive rewards for running along a straight path with a velocity limit for safety and stability.

**Performance and Constraint.** Figure 3 shows the learning curves for MICE and baseline methods in Safety Gym. The first row represents the cumulative discount reward of the episode during the training process. The second row is the cumulative discount cost, with the black dashed line indicating the cost threshold. The results indicate that MICE significantly reduces constraint violations during training while maintaining similar policy performance compared to baselines. Notably, in navigation tasks like PointGoal, the intrinsic cost provides predictive signals to avoid obstacles, allowing the policy performance of MICE-CPO to converge faster than the baseline CPO. In Safety MuJoCo, as shown in Figure 4, MICE achieves zero violation for the velocity constraint during training, with a convergence speed comparable to baselines. Our approach matches the constraint satisfaction levels of Saute and SimmerPID, which emphasize zero constraint violations, while surpassing their policy performance. Extended experiments covering a broader range of task types and robot types are provided in Appendix C.2.5.

**Mitigating Underestimation with MICE.** To assess the effectiveness of the extrinsic-intrinsic cost value function in MICE for mitigating underestimation, we compare the gap between the cost value

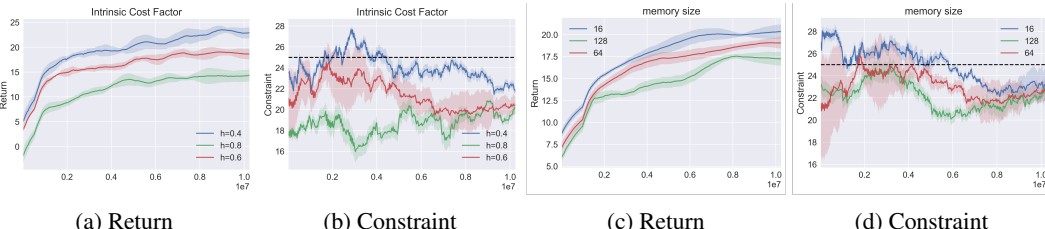

(a) Return       (b) Constraint       (c) Return       (d) Constraint

Figure 7: Sensitivity analysis of hyperparameters in MICE. **(a)(b)** Comparative experiment with different intrinsic factor $h$. **(c)(d)** Comparative experiment with different memory capacity.

estimates and their true values across MICE and baselines. True values are derived using the average discount constraint over 1,000 episodes under the current policy. Results in Figure 5a and 5b illustrate that the MICE method significantly mitigates the underestimation by enhancing the cost value estimates. Furthermore, the extrinsic-intrinsic cost value function gradually converges to the true value, confirming the convergence analysis in Theorem 3. Besides, we compared the TD3-based cost value function and MICE in mitigating constraint bias, detailed in Appendix C.2.4.

**Ablation Study of Intrinsic Cost.** To validate the effectiveness of memory-driven intrinsic cost estimate module in MICE, we construct comparison experiments between the MICE algorithm and versions that directly add various constants (5, 10, 15) into the cost value function. Results shown in Figure 6 demonstrate that adding constants decreases policy performance. Our method is both theoretically and empirically validated to converge to the optimal value. While this is not guaranteed in versions of adding constants, leading to reduced performance and sub-optimal results for the final policy. Compare to constants, the intrinsic cost signal contains more memory-related and task-related information, helping to avoid hazards and improve performance.

**Robustness to Constraint Thresholds.** To evaluate the adaptability of MICE to varying constraint thresholds, we construct sensitivity analysis experiments in SafetyPointGoal1-v0 with thresholds set at 0, 15, and 25, as illustrated in Figure 5c and 5d. The results show that MICE effectively accommodates different constraint threshold requirements. Specifically, when the threshold is set at 15, MICE balances policy performance with constraint satisfaction. In scenarios with a strict threshold of 0, MICE successfully achieves the policy striving to meet the constraints.

**Sensitivity Analysis of Hyperparameters.** The intrinsic factor and memory capacity are critical hyperparameters in MICE, and we conducted experiments to assess their sensitivity individually. (1) By modifying the intrinsic factor $h$, we can effectively adjust the agent's risk preference to suit various task requirements. Figure 7a and 7b demonstrate the robustness of MICE for different risk preferences in SafetyCarCircle-v0. Specifically, for tasks demanding high security, increasing $h$ effectively enhances the agent's risk aversion. While excessive risk aversion may compromise policy performance, it is essential for security-oriented tasks. $h$ is set to 0.6 in our work. More detailed analysis and experiments on the intrinsic factor are provided in Appendix C.2.1. (2) Memory capacity presents less impact on MICE performance compared to the intrinsic factor, as shown in Figure 7c and 7d in SafetyCarCircle-v0. A larger memory capacity slightly reduces constraint violation by allowing the storage of earlier trajectories that assist the agent in avoiding hazards. However, it may also retain less relevant trajectories, resulting in a more conservative policy. A smaller memory capacity causes a more aggressive policy due to the limited number of saved trajectories, which may not provide sufficient intrinsic signals when exploring new regions, potentially leading to constraint violations. We set the capacity as 64 in this paper. Detailed analysis is provided in Appendix C.2.2.

## 6 CONCLUSION

This paper highlights an important challenge in CRL, underestimation of the cost value, which significantly contributes to constraint violations. To mitigate the underestimation, we propose the MICE algorithm, which incorporates an extrinsic-intrinsic cost value update mechanism inspired by human cognitive processes. MICE enhances the cost estimates of unsafe trajectories, reducing the likelihood of constraint violations. Theoretically, we give the upper bound of constraint violations and convergence guarantees of the MICE algorithm. Extensive experimental results show that MICE reduces constraint violations while maintaining robust policy performance.

**Reproducibility Statement.** We make a lot of efforts to ensure reproducibility of our work. We provide a link to a anonymous downloadable source code in Appendix C.1. For the theoretical results presented in this paper, we provide clear explanations of all assumptions, along with complete proofs of the claims, which can be found in Appendix B. For the environments used in the experiments, a complete description of the tasks and agents included in environments is provided in Appendix C.3.

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

# A SAFETY POLICY OPTIMIZATION IN MICE

## A.1 NOTATIONS

**Notations**

| | |
|---|---|
| $c^I$ | intrinsic cost |
| $c^E$ | extrinsic cost |
| $R$ | reward function |
| $P$ | transition probability function |
| $\rho$ | initial state distribution |
| $\gamma$ | discount factor of the reward |
| $\pi$ | the policy |
| $d^\pi$ | the discounted future state visitation distribution |
| $\tau$ | trajectory |
| $C$ | extrinsic cost function |
| $Q_R^\pi(s,a)$ | action-value function |
| $V_R^\pi(s)$ | value function |
| $A_R^\pi(s,a)$ | advantage function |
| $J_R$ | the expected discount return |
| $J_C$ | the expected discount cost return |
| $d$ | cost threshold |
| $\Pi_C$ | the set of feasible policies |
| $Q_C^\pi(s,a)$ | action cost value function |
| $V_C^\pi(s)$ | cost value function |
| $A_C^\pi(s,a)$ | cost advantage function |
| $\epsilon$ | the noise in value estimate |
| $\hat{Q}_C(s,a)$ | the approximate cost value function |
| $\alpha$ | step size |
| $M$ | flashbulb memory |
| $\tau^m$ | unsafe trajectory |
| $G$ | intrinsic generator |
| $\phi$ | network parameters of intrinsic generator |
| $h$ | intrinsic factor |
| $\gamma_I$ | intrinsic discount factor |
| $W$ | weight vector of state-action pairs |
| $\omega$ | intrinsic weight |
| $Q_C^{EI}$ | extrinsic-intrinsic cost value function |
| $J_C^{EI}$ | cumulative discount extrinsic-intrinsic cost |
| $C^{EI}$ | extrinsic-intrinsic cost function |
| $A_C^{EI}$ | extrinsic-intrinsic cost advantage function |

The CRL aims to find an optimal policy by maximizing the expected discount return over the set of feasible policies $\Pi_C := \{\pi \in \Pi : J_C(\pi) \leq d\}$:

$$\arg \max_{\pi \in \Pi} J_R(\pi)$$
$$s.t. \quad J_C(\pi) \leq d \tag{16}$$

The following equation briefly gives the performance difference of arbitrary two policies, which represents the expected return of another policy $\pi'$ in terms of the advantage function over $\pi$:

$$J_R(\pi') - J_R(\pi) = \frac{1}{1-\gamma} \mathbb{E}_{s \sim d^{\pi'}, a \sim \pi'}[A_R^{\pi}(s,a)] \tag{17}$$

This implies that iterative updates to the policy, $\pi'(s) = \arg \max_a A_R^{\pi}(s,a)$, lead to performance improvement until convergence to the optimal solution.

According to the performance difference equation (17), CRL is defined as a constrained optimization problem:

$$\pi_{k+1} = \arg \max_{\pi \in \Pi_\theta} \mathbb{E}_{s \sim d^{\pi}, a \sim \pi}[A_R^{\pi_k}(s,a)]$$
$$s.t. \quad J_C(\pi_k) + \frac{1}{1-\gamma} \mathbb{E}_{s \sim d^{\pi}, a \sim \pi}[A_C^{\pi_k}(s,a)] \leq d \tag{18}$$

where policy $\pi \in \Pi_\theta$ is parameterized with parameters $\theta$, and $\pi_k$ represents the current policy.

In this paper, we define the cumulative discount extrinsic-intrinsic cost as $J_C^{EI}(\pi) := \mathbb{E}_{\tau \sim \pi}[\sum_{t=0}^{\infty} \gamma^t C^{EI}(s_t, a_t)] = \mathbb{E}_{\tau \sim \pi}[\sum_{t=0}^{\infty} \gamma^t(c^E + c^I)]$, where $C^{EI}(s,a) = c^E + c^I$ is the extrinsic-intrinsic cost function. The extrinsic-intrinsic advantage function in MICE is defined as:

$$A_C^{EI}(s,a) = \mathbb{E}_{s'}[c^E + c^I + \gamma V_C(s') - V_C(s)] \tag{19}$$

To reduce constraint violations, we replace $J_C$ with the extrinsic-intrinsic constraint $J_C^{EI}$ in the optimization objective. We give the optimization objective of MICE based on the extrinsic-intrinsic cost value estimate and Lemma 1:

$$\pi_{k+1} = \arg \max_{\pi \in \Pi_\theta} \mathbb{E}_{s \sim d^{\pi}, a \sim \pi}[A_R^{\pi_k}(s,a)]$$
$$s.t. \quad J_C(\pi_k) + \frac{1}{1-\gamma} \mathbb{E}_{s \sim d^{\pi}, a \sim \pi}[A_C^{EI}(s,a|\pi_k)] \leq d \tag{20}$$

where policy $\pi \in \Pi_\theta$ is parameterized with parameters $\theta$, and $\pi_k$ represents the current policy. We propose two optimization methods, MICE-CPO and MICE-PIDLag, based on CPO and PID Lagrangian respectively, to solve the optimization 20.

## A.2 MICE-CPO

The complex dependency of state visitation distribution $d^{\pi}(s)$ on unknown policy $\pi$ makes 20 difficult to optimize directly. To address this, this paper uses samples generated by the current policy $\pi_k$ to approximate the original problem locally. Based on the extrinsic-intrinsic value update function, we seek to solve the following optimization problem in the trust region:

$$\pi_{k+1} = \arg \max_{\pi \in \Pi_\theta} \mathbb{E}_{s \sim d^{\pi_k}, a \sim \pi}[A_R^{\pi_k}(s,a)]$$
$$s.t. \quad J_C(\pi_k) + \frac{1}{1-\gamma} \mathbb{E}_{s \sim d^{\pi_k}, a \sim \pi}[A_C^{EI}(s,a|\pi_k)] \leq d \tag{21}$$
$$D(\pi \| \pi_k) \leq \delta$$

where $\Pi_\theta$ is the policy set parameterized by parameter $\theta$, $D(\pi \| \pi_k) = \mathbb{E}_{s \sim d^{\pi_k}}[D_{KL(\pi \| \pi_k)}[s]]$, $D_{KL}$ is the KL divergence and $\delta > 0$ is the step size. The set $\{\pi \in \Pi_\theta : D(\pi \| \pi_k) \leq \delta\}$ is the trust region.

In the MICE-CPO method, we approximate the reward objective and cost constraints with first-order expansion and approximate the KL-divergence constraint with second-order expansion. The local approximation to 21 is:

$$
\begin{aligned}
\theta_{k+1} = \arg\max_{\theta} \; & g^T(\theta - \theta_k) \\
s.t. \quad & c + (g_C^{EI})^T(\theta - \theta_k) \leq 0 \\
& \frac{1}{2}(\theta - \theta_k)^T H(\theta - \theta_k) \leq \delta
\end{aligned}
\tag{22}
$$

where $g$ denotes the gradient of the reward objective in 21, $g_C^{EI}$ denotes the gradient of extrinsic-intrinsic constraint in 21, $c = J_C(\pi_k) - d$, $H$ is the Hessian of the KL-divergence. When the constraint is satisfied, we can get the analytical solution with the primal-dual method. The solution to the primal problem is

$$
\theta^* = \theta_k + \frac{1}{\lambda^*} H^{-1}(g - g_C^{EI}\nu^*)
\tag{23}
$$

where $\lambda$ and $\nu$ are the Lagrangian multipliers of the KL-divergence term and the constraint term in the Lagrangian function, respectively. $\lambda^*$, $\nu^*$ are the solutions to the dual problem:

$$
\nu^* = \max\{0, \frac{\lambda^* c - u}{v}\}
\tag{24}
$$

$$
\lambda^* = \arg\max_{\lambda \geq 0}
\begin{cases}
\frac{1}{2\lambda}\left(\frac{u^2}{v} - q\right) + \frac{\lambda}{2}\left(\frac{c^2}{v} - \delta\right) - \frac{uc}{v}, & \text{if}\lambda c > u \\
-\frac{1}{2}\left(\frac{q}{\lambda} + \lambda\delta\right), \text{otherwise,}
\end{cases}
\tag{25}
$$

where $q = g^T H^{-1} g$, $u = g^T H^{-1} g_C^{EI}$, $v = (g_C^{EI})^T H^{-1} g_C^{EI}$.

When the constraint is violated, we use the conjugate gradient method to decrease the constraint value:

$$
\theta^* = \theta_k - \left(\frac{2\delta}{(g_C^{EI})^T H^{-1} g_C^{EI}}\right)^{\frac{1}{2}} H^{-1} g_C^{EI}
\tag{26}
$$

### A.3 MICE-PIDLAG

In the MICE-PIDLag method, we write the CRL problem 20 as the first-order dynamical system:

$$
\begin{aligned}
\theta_{k+1} &= \theta_k + \eta(g - \lambda_k g_C^{EI}) \\
y_k &= J_C(\pi_k) + \frac{1}{1-\gamma}\mathbb{E}_{s\sim d^\pi, a\sim\pi}[A_C^{EI}(s, a|\pi_k)] \\
\lambda_k &= h(y_0, \cdots, y_k, d)
\end{aligned}
\tag{27}
$$

where $\eta$ is the step size of the update, $g$ is the gradient of reward objective in 20 and $g_C^{EI}$ denotes the gradient of extrinsic-intrinsic constraint in 20. $h$ denotes the control function. $\lambda$ is the Lagrangian multiplier for the 20. We provide the updated formulas for the Lagrangian multiplier in MICE-PIDLag:

$$
\lambda \leftarrow (K_P\Delta + K_I I + K_D\partial)_+
\tag{28}
$$

where $(\cdot)_+ = \max\{0, \cdot\}$, and

$$\Delta \leftarrow (J_C(\pi_k) + \frac{1}{1-\gamma}\mathbb{E}_{s\sim d^\pi, a\sim\pi}[A_C^{EI}(s, a|\pi_k)] - d),$$

$$I \leftarrow (I + \Delta)_+,$$

$$\partial \leftarrow \left(J_C(\pi_k) + \frac{1}{1-\gamma}\mathbb{E}_{s\sim d^\pi, a\sim\pi}[A_C^{EI}(s, a|\pi_k)] - J_C(\pi_{k-1}) - \frac{1}{1-\gamma}\mathbb{E}_{s\sim d^\pi, a\sim\pi}[A_C^{EI}(s, a|\pi_{k-1})]\right)_+$$

$$= \frac{1}{1-\gamma}\left(\mathbb{E}_{s\sim d^\pi, a\sim\pi}[A_C^{EI}(s, a|\pi_k)] - \mathbb{E}_{s\sim d^\pi, a\sim\pi}[A_C^{EI}(s, a|\pi_{k-1})] + \mathbb{E}_{s\sim d^{\pi_k}, a\sim\pi_k}[A_C^{\pi_{k-1}}(s, a)]\right)_+$$

(29)

$K_P$, $K_I$, and $K_D$ are the coefficients of the respective control terms. The initial value of the integral term $I$ is 0.

# B  THEORETICAL PROOF

In this section, we provide theoretical guarantees for our approach from several perspectives. First, we present the estimation bias lemma and the expected constraint difference between arbitrary two policies under the new cost value function. Then, we provide an extrinsic-intrinsic constraint bound and a tighter constraint violation upper bound in the MICE update. Additionally, we demonstrate that our cost value function can converge to the optimal solution.

## B.1  ESTIMATION BIAS LEMMA

**Lemma 2.** *In a finite MDP for a given state-action pair $(s, a)$, the difference between the optimal cost value function $Q_C^*(s, a)$ and the cost value estimate $Q_{C,m}^{EI}(s, a)$ after $m$ updates is given by:*

$$Q_C^*(s, a) - Q_{C,n+m}^{EI}(s, a) = (1-\alpha)^m[Q_C^*(s, a) - Q_{C,n}^{EI}(s, a)] - \alpha\sum_{i=1}^{m}(1-\alpha)^{i-1}t_{n+m-i}(s, a) \quad (30)$$

*where $Q_{C,n}^{EI}(s, a)$ is the estimate of the value function at the $n - th$ update, $\alpha$ is the step size, and $t_n(s, a) = c^E + c^I + \gamma\min_{a'}\mathbb{E}_{s'}[Q_{C_n}^{EI}(s', a')] - Q_C^*(s, a)$ is the target difference.*

*Proof.* We use induction method to proof this lemma 2.

Base Case: m = 1

Substituting $m = 1$ in lemma 2, we get:

$$Q_C^*(s, a) - Q_{C,n+1}^{EI}(s, a) = (1-\alpha)[Q_C^*(s, a) - Q_{C,n}^{EI}(s, a)] - \alpha t_n(s, a) \quad (31)$$

According to the update equation of the extrinsic-intrinsic cost value function in MICE, we get:

$$Q_{C,n+1}^{EI}(s, a) = (1-\alpha)Q_{C,n}^{EI}(s, a) + \alpha(c^E + c^I + \gamma\min_{a'}\mathbb{E}_{s'}[Q_{C,n}^{EI}(s', a')])$$
$$= (1-\alpha)Q_{C,n}^{EI}(s, a) + \alpha(t_n(s, a) + Q_C^*(s, a)) \quad (32)$$

which is equivalent to equation 31.

Induction Step: $m = k + 1$

Assuming lemma 2 is true for $m = k$, which is:

$$Q_C^*(s, a) - Q_{C,n+k}^{EI}(s, a) = (1-\alpha)^k[Q_C^*(s, a) - Q_{C,n}^{EI}(s, a)] - \alpha\sum_{i=1}^{k}(1-\alpha)^{i-1}t_{n+k-i}(s, a) \quad (33)$$

Now we need to prove that it holds for $m = k + 1$. According to the cost value update equation in MICE, we get:

$$Q^{EI}_{C,n+k+1}(s,a) = (1-\alpha)Q^{EI}_{C,n+k}(s,a) + \alpha(c^E + c^I + \gamma \min_{a'} \mathbb{E}_{s'}[Q^{EI}_{C,n+k}(s',a')])$$
$$= (1-\alpha)Q^{EI}_{C,n+k}(s,a) + \alpha(t_{n+k}(s,a) + Q^*_C(s,a)) \quad (34)$$

Then we can get:

$$Q^*_C(s,a) - Q^{EI}_{C,n+k+1}(s,a) = (1-\alpha)[Q^*_C(s,a) - Q^{EI}_{C,n+k}(s,a)] - \alpha t_{n+k}(s,a) \quad (35)$$

Substituting the equation 33, we get:

$$Q^*_C(s,a) - Q^{EI}_{C,n+k+1}(s,a)$$
$$=(1-\alpha)\left[(1-\alpha)^k[Q^*_C(s,a) - Q^{EI}_{C,n}(s,a)] - \alpha \sum_{i=1}^{k}(1-\alpha)^{i-1}t_{n+k-i}(s,a)\right] - \alpha t_{n+k}(s,a)$$
$$=(1-\alpha)^{k+1}[Q^*_C(s,a) - Q^{EI}_{C,n}(s,a)] - \alpha \sum_{i=1}^{k+1}(1-\alpha)^{i-1}t_{n+k+1-i}(s,a)$$
$$(36)$$

which satisfies the equation when $m = k + 1$ in lemma 2.

$\square$

Lemma 2 indicates that when $n = 0$ and $Q^{EI}_{C,n}(s,a)$ is a random initial value for the cost value function, in a stochastic high value region of the state-action space, it is likely that $Q^*_C(s,a) > Q^{EI}_{C,n}(s,a)$ Karimpanal et al. (2023). In this case, overestimation of our extrinsic-intrinsic cost value function can effectively reduce the estimation bias, whereas underestimation in the traditional value function further increases the estimation bias.

## B.2 Constraint Difference Lemma

**Lemma 3.** *Given arbitrary two policies $\pi$ and $\pi'$, the difference in expectation constraint of extrinsic-intrinsic $J^{EI}_C(\pi')$ and extrinsic $J_C(\pi)$ can be expressed as:*

$$J^{EI}_C(\pi') - J_C(\pi) = \mathbb{E}_{\tau|\pi'}\left[\sum_{t=0}^{\infty}\gamma^t A^{EI}_C(s_t, a_t|\pi)\right] \quad (37)$$

*where $A^{EI}_C(s_t, a_t|\pi) = \mathbb{E}_{s_{t+1}}[c^E_t + c^I_t + \gamma V^\pi_C(s_{t+1}) - V^\pi_C(s_t)]$. The expectation is taken over trajectories $\tau$, and $\mathbb{E}_{\tau|\pi'}$ indicates that actions are sampled from $\pi'$ to generate $\tau$.*

*Proof.* The expectations in $J^{EI}_C(\pi')$ and $J_C(\pi)$ can be expanded as:

$$J^{EI}_C(\pi') := \mathbb{E}_{\tau\sim\pi'}[\sum_{t=0}^{\infty}\gamma^t(c^E(s_t, \pi'(s_t)) + c^I(s_t, \pi'(s_t)))]$$
$$J_C(\pi) := \mathbb{E}_{\tau\sim\pi}[\sum_{t=0}^{\infty}\gamma^t c^E(s_t, \pi(s_t))] = \mathbb{E}_{s_0\sim\rho}[V^\pi_C(s_0)] \quad (38)$$

$$J_C^{EI}(\pi') - J_C(\pi)$$

$$=\mathbb{E}_{\tau|\pi'}\left[\sum_{t=0}^{\infty}\gamma^t\left(c^E(s_t,\pi'(s_t)) + c^I(s_t,\pi'(s_t))\right)\right] - \mathbb{E}_{s_0\sim\rho}[V_C^{\pi}(s_0)]$$

$$=\mathbb{E}_{\tau|\pi'}\left[\sum_{t=0}^{\infty}\gamma^t\left(c^E(s_t,\pi'(s_t)) + c^I(s_t,\pi'(s_t))\right) - V_C^{\pi}(s_0)\right]$$

$$=\mathbb{E}_{\tau|\pi'}\left[\sum_{t=0}^{\infty}\gamma^t\left(c^E(s_t,\pi'(s_t)) + c^I(s_t,\pi'(s_t) + V_C^{\pi}(s_t) - V_C^{\pi}(s_t))\right) - V_C^{\pi}(s_0)\right] \quad (39)$$

$$=\mathbb{E}_{\tau|\pi'}[-V_C^{\pi}(s_0) + c^E(s_0,\pi'(s_0)) + c^I(s_0,\pi'(s_0)) + V_C^{\pi}(s_0) - V_C^{\pi}(s_0)$$
$$+ \gamma c^E(s_1,\pi'(s_1)) + \gamma c^I(s_1,\pi'(s_1)) + \gamma V_C^{\pi}(s_1) - \gamma V_C^{\pi}(s_1) + \cdots]$$

$$=\mathbb{E}_{\tau|\pi'}\left[\sum_{t=0}^{\infty}\gamma^t\left(c^E(s_t,\pi'(s_t)) + c^I(s_t,\pi'(s_t)) + \gamma V_C^{\pi}(s_{t+1}) - V_C^{\pi}(s_t)\right)\right]$$

$$=\mathbb{E}_{\tau|\pi'}\left[\sum_{t=0}^{\infty}\gamma^t A_C^{EI}(s_t,\pi'(s_t)|\pi)\right]$$

Here the term $A_C^{EI}(s_t,\pi'(s_t)|\pi)$ denotes that the advantage value function $A_C^{EI}$ is over $\pi$, the action is selected according to $\pi'$.

The second equation above holds because that

$$\mathbb{E}_{\tau|\pi'}\left[V_C^{\pi}(s_0)\right]$$
$$=\mathbb{E}_{s\sim d^{\pi'},a\sim\pi',s'\sim P}\left[V_C^{\pi}(s_0)\right] \quad (40)$$
$$=\mathbb{E}_{s_0\sim\rho}\left[V_C^{\pi}(s_0)\right]$$

The initial state $s_0$ in $V_C^{\pi}(s_0)$ depends solely on the initial state distribution $\rho$, allowing the expectation over $\tau|\pi'$ to be expressed as an expectation over $s_0 \sim \rho$.

The third equation in proof holds by adding $V_C^{\pi}$ while subtracting $V_C^{\pi}$. The fourth equation expands the cumulative sum over time steps $t$. The final equation follows from the definition of $A_C^{EI}$.

□

The performance difference theorem is a fundamental property in RL that describes the relationship between the difference in expected cumulative rewards of arbitrary two policies and the advantage function. Similarly, we provide an expression for the difference between the expected constraints of two policies based on the extrinsic-intrinsic cost value function.

**Lemma 4.** *Given arbitrary two policy $\pi$ and $\pi'$, the difference in expectation of cumulative cost can be expressed as:*

$$J_C(\pi') - J_C(\pi) = \mathbb{E}_{\tau|\pi'}\left[\sum_{t=0}^{\infty}\gamma^t A_C^{EI}(s_t,a_t|\pi)\right] - I \quad (41)$$

*where $A_C^{EI}(s_t,a_t|\pi) = \mathbb{E}_{s_{t+1}}[c_t^E + c_t^I + \gamma V_C^{\pi}(s_{t+1}) - V_C^{\pi}(s_t)]$, $I = \mathbb{E}_{\tau|\pi'}\left[\sum_{t=0}^{\infty}\gamma^t c_t^I\right]$.. The expectation is taken over trajectories $\tau$, and $\mathbb{E}_{\tau|\pi'}$ indicates that actions are sampled from $\pi'$ to generate $\tau$.*

*Proof.*

$$\mathbb{E}_{\tau|\pi'}\left[\sum_{t=0}^{\infty}\gamma^t A_C^{EI}(s_t, a_t|\pi)\right]$$

$$=\mathbb{E}_{\tau|\pi'}\left[\sum_{t=0}^{\infty}\gamma^t\left(c_t^E + c_t^I + \gamma V^\pi(s_{t+1}) - V^\pi(s_t)\right)\right]$$

$$=\mathbb{E}_{\tau|\pi'}\left[\sum_{t=0}^{\infty}\gamma^t\left(c_t^E + c_t^I\right) - V^\pi(s_0)\right] \tag{42}$$

$$=\mathbb{E}_{\tau|\pi'}\left[\sum_{t=0}^{\infty}\gamma^t\left(c_t^E + c_t^I\right)\right] - \mathbb{E}_{s_0}[V^\pi(s_0)]$$

$$=J_C(\pi') + \mathbb{E}_{\tau|\pi'}\left[\sum_{t=0}^{\infty}\gamma^t c_t^I\right] - J_C(\pi)$$

$\square$

### B.3 Constraint Bounds

For extrinsic-intrinsic constraints in MICE, we give an upper bound on the constraint difference:

**Theorem 4** (Extrinsic-intrinsic Constraint Bounds). *For arbitrary two policies $\pi'$ and $\pi$, the following bound for cumulative discount extrinsic-intrinsic cost holds:*

$$J_C^{EI}(\pi') - J_C^{EI}(\pi) \le \frac{1}{1-\gamma}\mathbb{E}_{s\sim d^\pi, a\sim \pi'}\left[A_C^{EI}(s,a|\pi) + \frac{2\gamma\epsilon_{\pi'}^{EI}}{1-\gamma}D_{TV}(\pi'\|\pi)[s]\right] \tag{43}$$

*where $\epsilon_{\pi'}^{EI} := \max_s |\mathbb{E}_{a\sim\pi', s'\sim P}[\varepsilon_V^{EI}(s,a,s')]|$, $\varepsilon_V^{EI}(s,a,s') = C^{EI}(s,a,s') + \gamma V_C(s') - V_C(s)$ denotes the extrinsic-intrinsic TD-error, $D_{TV}(\pi'\|\pi)[s] = (1/2)\sum_a |\pi'(a|s) - \pi(a|s)|$.*

The upper bound in Theorem 4 is related to the TV divergence $D_{TV}(\pi'\|\pi)[s]$ between $\pi$ and $\pi'$. $D_{TV}(\pi'\|\pi)$ is the total variation divergence, as mentioned in TRPO and CPO, which is:

$$D_{TV}(\pi'\|\pi)[s] = \frac{1}{2}\sum_a |\pi'(a|s) - \pi(a|s)| \tag{44}$$

A larger divergence between the two policies results in a larger upper bound on the constraint gap. This relationship supports the optimization objective 21 within the trust region in MICE.

*Proof.* Define the state visit probability for time step $t$ as $p_\pi^t(s) = P(s_t = s|\pi)$, denote the transition matrix as $P_\pi(s'|s) = \int da\pi(a|s)P(s'|s,a)$, we get $p_\pi^t = P_\pi p_\pi^{t-1} = \cdots = P_\pi^t\rho$. The discounted future state distribution $d^\pi(s)$ satisfies:

$$d^\pi(s) = (1-\gamma)\sum_{t=0}^{\infty}\gamma^t P(s_t = s|\pi)$$

$$= (1-\gamma)\sum_{t=0}^{\infty}\gamma^t p_\pi^t(s)$$

$$= (1-\gamma)\sum_{t=0}^{\infty}\gamma^t P_\pi^t\rho \tag{45}$$

$$= (1-\gamma)\sum_{t=0}^{\infty}(\gamma P_\pi)^t\rho$$

$$= (1-\gamma)(I - \gamma P_\pi)^{-1}\rho$$

where $\rho$ is the initial state distribution, $I$ is the identity matrix. Multiply both sides by $(I - \gamma P_\pi)$, we get

$$(I - \gamma P_\pi)d^\pi(s) = (1 - \gamma)\rho \tag{46}$$

For cost value function $V_C(s)$ with polices $\pi'$ and $\pi$, we get the following according to the 46:

$$
\begin{aligned}
&(1 - \gamma)\mathbb{E}_{s\sim\rho}[V_C(s)] + \mathbb{E}_{s\sim d^\pi, a\sim\pi, s'\sim P}[\gamma V_C(s')] - \mathbb{E}_{s\sim d^\pi}[V_C(s)] \\
=&(1 - \gamma)\int \mathrm{d}s\rho(s)V_C(s) + \int \mathrm{d}s \int \mathrm{d}a \int \mathrm{d}s' d^\pi(s)\pi(a|s)P(s'|s,a)\gamma V_C(s') - \int \mathrm{d}s d^\pi(s)V_C(s) \\
=&\int \mathrm{d}s(1 - \gamma)\rho(s)V_C(s) + \int \mathrm{d}s d^\pi(s)P_\pi\gamma V_C(s') - \int \mathrm{d}s d^\pi(s)V_C(s) \\
=&\int \mathrm{d}s(I - \gamma P_\pi)d^\pi(s)V_C(s) + \int \mathrm{d}s d^\pi(s)P_\pi\gamma V_C(s') - \int \mathrm{d}s d^\pi(s)V_C(s) \\
=&0
\end{aligned}
\tag{47}
$$

The third equation above holds as the 46. Then we get:

$$(1 - \gamma)\mathbb{E}_{s\sim\rho}[V_C(s)] + \mathbb{E}_{s\sim d^\pi, a\sim\pi, s'\sim P}[\gamma V_C(s')] - \mathbb{E}_{s\sim d^\pi}[V_C(s)] = 0 \tag{48}$$

The definition of discount total extrinsic-intrinsic cost is:

$$J_C^{EI}(\pi) = \frac{1}{1 - \gamma}\mathbb{E}_{s\sim d^\pi, a\sim\pi, s'\sim P}[C^{EI}(s, a, s')] \tag{49}$$

By combining this with 48, we get the discount total extrinsic-intrinsic cost equation:

$$J_C^{EI}(\pi) = \mathbb{E}_{s\sim\rho}[V_C(s)] + \frac{1}{1 - \gamma}\mathbb{E}_{s\sim d^\pi, a\sim\pi, s'\sim P}[C^{EI}(s, a, s') + \gamma V_C(s') - V_C(s)] \tag{50}$$

where the first term on the right side is the estimate of the policy constraint, and the second term on the right side is the average extrinsic-intrinsic TD-error of the approximator.

The extrinsic-intrinsic TD-error $\varepsilon_V^{EI}(s, a, s') = C^{EI}(s, a, s') + \gamma V_C(s') - V_C(s)$. According to the equation 50, the expectation extrinsic-intrinsic constraint difference of any two policies is:

$$J_C^{EI}(\pi') - J_C^{EI}(\pi) = \frac{1}{1 - \gamma}\left(\mathbb{E}_{s\sim d^{\pi'}, a\sim\pi', s'\sim P}[\varepsilon_V^{EI}(s, a, s')] - \mathbb{E}_{s\sim d^\pi, a\sim\pi, s'\sim P}[\varepsilon_V^{EI}(s, a, s')]\right) \tag{51}$$

To simplify the representation, we denote $\bar{\varepsilon}_{\pi'}(s) = \mathbb{E}_{a\sim\pi', s'\sim P}[\varepsilon_V^{EI}(s, a, s')]$. The first term of the right side in 51 can be represented as:

$$
\begin{aligned}
\mathbb{E}_{s\sim d^{\pi'}, a\sim\pi', s'\sim P}[\varepsilon_V^{EI}(s, a, s')] &= \int \mathrm{d}s d^{\pi'} \int \mathrm{d}a\pi' \int \mathrm{d}s' P \varepsilon_V^{EI}(s, a, s') \\
&= \langle d^{\pi'}, \bar{\varepsilon}_{\pi'}\rangle \\
&= \langle d^\pi, \bar{\varepsilon}_{\pi'}\rangle + \langle d^{\pi'} - d^\pi, \bar{\varepsilon}_{\pi'}\rangle
\end{aligned}
\tag{52}
$$

the second equation holds by adding $d^\pi$ while subtracting $d^\pi$.

According to the Hölder's inequality, for any $p, q \in [1, \infty]$ satisfy $\frac{1}{p} + \frac{1}{q} = 1$, we set $p = 1$ and $q = \infty$, and get:

$$\langle d^{\pi'} - d^\pi, \bar{\varepsilon}_{\pi'}\rangle \le \|d^{\pi'} - d^\pi\|_1 \|\bar{\varepsilon}_{\pi'}\|_\infty \tag{53}$$

According to the definition in this theorem, we have $\|\bar{\varepsilon}_{\pi'}\|_\infty = \epsilon_{\pi'}^{EI}$, and $\|d^{\pi'} - d^\pi\|_1 = 2D_{TV}(d^{\pi'}\|d^\pi)$. According to the Lemma 3 in CPO, we have:

$$\|d^{\pi'} - d^\pi\|_1 \le \frac{2\gamma}{1-\gamma}\mathbb{E}_{s\sim d^\pi}[D_{TV}(\pi'\|\pi)[s]] \tag{54}$$

By the importance sampling, we get:

$$\langle d^\pi, \bar{\varepsilon}_{\pi'}\rangle = \langle\frac{\pi'}{\pi}d^\pi, \bar{\varepsilon}_\pi\rangle \tag{55}$$

The second term of the right side in 51 can be represented as:

$$\mathbb{E}_{s\sim d^\pi, a\sim\pi, s'\sim P}[\varepsilon_V^{EI}(s, a, s')] = \int \mathrm{d}s d^\pi \int \mathrm{d}a \pi \int \mathrm{d}s' P \varepsilon_V^{EI}(s, a, s')$$
$$= \langle d^\pi, \bar{\varepsilon}_\pi\rangle \tag{56}$$

Then we get the final result by combining the above equations:

$$J_C^{EI}(\pi') - J_C^{EI}(\pi) \le \frac{1}{1-\gamma}\left(\langle\frac{\pi'}{\pi}d^\pi, \bar{\varepsilon}_\pi\rangle + 2D_{TV}(d^{\pi'}\|d^\pi)\epsilon_{\pi'}^{EI} - \langle d^\pi, \bar{\varepsilon}_\pi\rangle\right)$$
$$= \frac{1}{1-\gamma}\left((\frac{\pi'}{\pi} - 1)\langle d^\pi, \bar{\varepsilon}_\pi\rangle + 2D_{TV}(d^{\pi'}\|d^\pi)\epsilon_{\pi'}^{EI}\right)$$
$$\le \frac{1}{1-\gamma}\mathbb{E}_{s\sim d^\pi, a\sim\pi, s'\sim P}\left[(\frac{\pi'}{\pi} - 1)\varepsilon_V^{EI}(s, a, s') + \frac{2\gamma\epsilon_{\pi'}^{EI}}{1-\gamma}D_{TV}(\pi'\|\pi)[s]\right]$$
$$= \frac{1}{1-\gamma}\mathbb{E}_{s\sim d^\pi, a\sim\pi'}\left[A_C^{EI}(s, a|\pi) + \frac{2\gamma\epsilon_{\pi'}^{EI}}{1-\gamma}D_{TV}(\pi'\|\pi)[s]\right]$$
$$\tag{57}$$

$$\square$$

By mitigating the underestimation, the MICE algorithm significantly reduces constraint violations during the learning process. We further establish a theoretical upper bound on the constraint violation for the updated policy within the optimization framework of MICE:

**Theorem 5** (MICE Update Worst-Case Constraint Violation). *Suppose $\pi_k$, $\pi_{k+1}$ are related by the optimization objective 21, an upper bound on the constraint of the updated policy $\pi_{k+1}$ is:*

$$J_C(\pi_{k+1}) \le d - I + \frac{\sqrt{2\delta}\gamma\epsilon_C^{\pi_{k+1}}}{(1-\gamma)^2} \tag{58}$$

*where $\epsilon_C^{\pi_{k+1}} := \max_s |\mathbb{E}_{a\sim\pi_{k+1}}[A_C^{\pi_k}(s, a)]|$, $I = \mathbb{E}_{\tau|\pi_{k+1}}\left[\sum_{t=0}^\infty \gamma^t c_t^I\right]$.*

Theorem 5 demonstrates that our method achieves a tighter upper bound on constraint violation compared to CPO, which guarantees that the updated policy in MICE has a lower probability of exceeding the constraint limits.

*Proof.* According to the Corollary 2 in CPO Achiam et al. (2017),

$$J_C(\pi_{k+1}) - J_C(\pi_k) \le \frac{1}{1-\gamma}\mathbb{E}_{s\sim d^{\pi_k}, a\sim\pi_{k+1}}\left[A_C^{\pi_k}(s, a) + \frac{2\gamma\epsilon_C^{\pi_{k+1}}}{1-\gamma}D_{TV}(\pi_{k+1}\|\pi_k)[s]\right] \tag{59}$$

As $\pi_k$, $\pi_{k+1}$ are related by 21, we get

$$J_C(\pi_k) + \frac{1}{1-\gamma}\mathbb{E}_{s\sim d^{\pi_k}, a\sim\pi_{k+1}}[A_C^{EI}(s, a|\pi_k)] \le d \tag{60}$$

which is:

$$J_C(\pi_k) + \frac{1}{1-\gamma}\mathbb{E}_{s\sim d^{\pi_k},a\sim\pi_{k+1}}[c^E + c^I + \gamma V_C(s') - V_C(s)] \le d$$

$$J_C(\pi_k) + \frac{1}{1-\gamma}\mathbb{E}_{s\sim d^{\pi_k},a\sim\pi_{k+1}}[A_C^{\pi_k}(s,a)] + \frac{1}{1-\gamma}\mathbb{E}_{s\sim d^{\pi_k},a\sim\pi_{k+1}}[c^I] \le d \tag{61}$$

$$J_C(\pi_k) + \frac{1}{1-\gamma}\mathbb{E}_{s\sim d^{\pi_k},a\sim\pi_{k+1}}[A_C^{\pi_k}(s,a)] \le d - I$$

According to Pinsker's inequality, for arbitrary distributions $p$, $q$, the TV-divergence and KL-divergence are related by:

$$D_{TV}(p||q) \le \sqrt{\frac{D_{KL}(p||q)}{2}} \tag{62}$$

According to Jensen's inequality, we get:

$$\mathbb{E}_{s\sim d^{\pi_k}}[D_{TV}(\pi_{k+1}||\pi_k)[s]] \le \sqrt{\frac{1}{2}\mathbb{E}_{s\sim d^{\pi_k}}[D_{KL}(\pi_{k+1}||\pi_k)[s]]}$$
$$\le \sqrt{\frac{\delta}{2}} \tag{63}$$

Then we get the final result:

$$J_C(\pi_{k+1}) \le d - I + \frac{\sqrt{2\delta}\gamma\epsilon_C^{\pi_{k+1}}}{(1-\gamma)^2} \tag{64}$$

$\square$

### B.4 CONVERGENCE ANALYSIS

Based on the same assumptions as in TD3 and Double Q-learning, we give convergence guarantees of the extrinsic-intrinsic cost value function in MICE.

**Lemma 5.** *Consider a stochastic process $(\zeta_t, \Delta_t, F_t), t \ge 0$ where $\zeta_t, \Delta_t, F_t : X \to \mathbb{R}$ satisfy the equation:*

$$\zeta_{t+1}(x_t) = (1 - \zeta_t(x_t))\Delta_t(x_t) + \zeta_t(x_t)F_t(x_t) \tag{65}$$

*where $x_t \in X$ ant $t = 0, 1, 2, \cdots$. Let $P_t$ be a sequence of increasing $\sigma$-fields such that $\zeta_0$ and $\Delta_0$ are $P_0$-measurable and $\zeta_t$, $\Delta_t$ and $F_{t-1}$ are $P_t$-measurable, $t = 1, 2, \cdots$. Assume that the following holds:*

1. *The set $X$ is finite.*

2. *$\zeta_t(x_t) \in [0, 1]$, $\sum_t \zeta_t(x_t) = \infty$, $\sum_t (\zeta_t(x_t))^2 < \infty$ with probability 1 and $\forall x \ne x_t : \zeta(x) = 0$.*

3. *$\| \mathbb{E}[F_t|P_t] \|_\infty \le \kappa \| \Delta_t \|_\infty + c_t$ where $\kappa \in [0, 1)$ and $c_t$ converges to 0 with probability 1.*

4. *$Var[F_t(x_t)|P_t] \le K(1 + \kappa \| \Delta_t \|_\infty)^2$, where $K$ is some constant.*

*where $\| \cdot \|_\infty$ denotes the maximum norm. Then $\Delta_t$ converges to 0 with probability 1.*

We use the Lemma 5 to prove the convergence of our approach with a similar condition in Q-learning.

**Theorem 6** (Convergence Analysis). *Given the following conditions:*

1. *Each state-action pair is sampled an infinite number of times.*

2. *The MDP is finite.*

3. $\gamma \in [0, 1)$.

4. $Q_C$ *values are stored in a lookup table.*

5. $Q_C$ *receives an infinite number of updates.*

6. *The learning rates satisfy $\alpha_t(s, a) \in [0, 1]$, $\sum_t \alpha_t(s, a) = \infty$, $\sum_t (\alpha_t(s, a))^2 < \infty$ with probability 1 and $\alpha_t(s, a) = 0, \forall(s, a) \neq (s_t, a_t)$.*

7. $Var[c_t^E + c_t^I] < \infty, \forall s, a$.

*The extrinsic-intrinsic $Q_C^{EI}$ will converge to the optimal value function $Q_C^*$ with probability 1.*

Theorem 6 ensures that our method converges to the optimal solution.

*Proof.* We apply Lemma 5 to prove Theorem 6. Denote the variables in Lemma 5 with $P_t = \{Q_{C0}^{EI}, s_0, a_0, \alpha_0, c_1^E, s_1, \cdots, s_t, a_t\}$, $X = S \times A$, $\zeta_t = \alpha_t$. Define $\Delta_t(s_t, a_t) = Q_{Ct}^{EI}(s_t, a_t) - Q_C^*(s_t, a_t)$, $F_t = c_t^E + c_t^I + \gamma Q_{Ct}^{EI}(s_{t+1}, a^*) - Q_C^*(s_t, a_t)$, where $a^* = \arg\min_a Q_C^{EI}(s_{t+1}, a)$.

Condition 1 of the lemma 5 holds by condition 2 of the theorem 6. Condition 2 of the lemma 5 holds as the theorem condition 6 with $\zeta_t = \alpha_t$. The condition 4 of lemma 5 holds as a consequence of the condition 7 in the theorem.

So we need to show that the lemma condition 3 on the expected contraction of $F_t$ holds.

The extrinsic-intrinsic Q-learning equation in our paper is:

$$Q_C^{EI}(s, a) = (1 - \alpha)Q_C^{EI}(s, a) + \alpha(c^E + c^I + \gamma \min_{a'} \mathbb{E}_{s'}[Q_C^{EI}(s', a')]) \tag{66}$$

We have

$$\begin{aligned} \Delta_{t+1}(s_t, a_t) &= Q_{Ct+1}^{EI}(s_t, a_t) - Q_C^*(s_t, a_t) \\ &= (1 - \alpha_t)Q_{Ct}^{EI}(s_t, a_t) + \alpha_t(c_t^E + c_t^I + \gamma Q_t^{EI}(s_{t+1}, a^*)) - Q_C^*(s_t, a_t) \\ &= (1 - \alpha_t)(Q_{Ct}^{EI}(s_t, a_t) - Q_C^*(s_t, a_t)) + \alpha_t(c_t^E + c_t^I + \gamma Q_{Ct}^{EI}(s_{t+1}, a^*) - Q_C^*(s_t, a_t)) \\ &= (1 - \alpha_t)\Delta_t + \alpha_t F_t \end{aligned} \tag{67}$$

For the $F_t$, we can write

$$\begin{aligned} F_t(s_t, a_t) &= c_t^E + c_t^I + \gamma Q_{Ct}^{EI}(s_{t+1}, a^*) - Q^*(s_t, a_t) \\ &= F_t^E(s_t, a_t) + c_t^I \end{aligned} \tag{68}$$

where $F_t^E(s_t, a_t) = c_t^E + \gamma Q_{Ct}^{EI}(s_{t+1}, a^*) - Q_C^*(s_t, a_t)$ is the value of $F_t$ in normal Q-learning. According to the convergence analysis in Q-learning, we get $\mathbb{E}[F_t^E|P_t] \leq \gamma \parallel \Delta_t \parallel_\infty$. Then condition 3 of lemma 2 holds if $c_t^I$ converges to 0 with probability 1.

The intrinsic cost in our paper is defined as:

$$c^I = \frac{h\gamma_I^k}{(1 + e^{l_2})} \tag{69}$$

where $\gamma_I \sim (0, 1)$. So $c^I$ converges to 0 with probability 1, which then shows condition 3 of lemma 5 is satisfied. So the $Q_C^{EI}(s_t, a_t)$ converges to $Q^*(s_t, a_t)$.

$\square$

## C EXPERIMENT

### C.1 ALGORITHM PROCESS

We provide the main code in an anonymized form for MICE-CPO and MICE-PIDLag in `https://anonymous.4open.science/r/ICLR25-6568`. A formal description of our method is shown in Algorithm 1.

---

**Algorithm 1** MICE: Memory-driven Intrinsic Cost Estimation

---

**Input:** Initialize policy network $\pi_\theta$, value networks $V_R^\omega$ and $V_C^\psi$, flashbulb memory $M$, and intrinsic generator $G_\phi$. Set the hyperparameter.
**Output:** The optimal policy parameter $\theta$.

1: **for** epoch k=0,1,2,... **do**
2:     Sample N trajectories $\tau_1, ..., \tau_N$ under the current policy $\pi_{\theta_k}$.
3:     Update flashbulb memory $M$.
4:     Output the intrinsic cost $c^I$ by the intrinsic generator.
5:     Process the trajectories to $C$-returns, calculate extrinsic-intrinsic advantage functions $A^{EI}$
       with $V_C^\psi$ and $c^I$ by GAE method Schulman et al. (2015).
6:     **for** K iterations **do**
7:        Update value networks $V_R^\omega$, $V_C^\psi$, and intrinsic generator $G_\phi$.
8:        Update policy network $\pi_\theta$.
9:        **if** $\frac{1}{N}\sum_{j=1}^{N} D_{KL}(\pi_\theta||\pi_{\theta_k})[s_j] > \delta$ **then**
10:         Break.
11:       **end if**
12:    **end for**
13: **end for**
14: **return** policy parameters $\theta = \theta_{k+1}$.

---

## C.2 ADDITIONAL EXPERIMENTS

We design comparative experiments to verify the effect of hyperparameters in MICE and ablation experiments to verify the effectiveness of the components in MICE. Additionally, we conduct comparative experiments against more baselines.

### C.2.1 INTRINSIC FACTOR

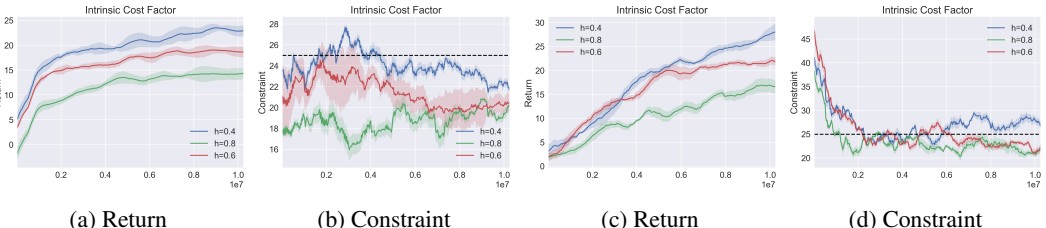

    (a) Return         (b) Constraint         (c) Return         (d) Constraint

Figure 8: Sensitivity analysis of MICE algorithm for different **intrinsic factor** in different environments. **(a)(b)** are in the SafetyCarCirlce1-v0 environment, **(c)(d)** are in the SafetyPointGoal1-v0 environment.

We construct experiments in different environments to analyze the sensitivity of the MICE algorithm to various intrinsic factors. The results are illustrated in Figure 8, where $h = 0.6$ is the value set in our paper.

Increasing the intrinsic factor enhances the intrinsic cost, thereby raising the agent's risk awareness. This adjustment can effectively mitigate the underestimation bias in cost value estimates and reduce constraint violations, which is particularly important for safety-critical tasks that require a more conservative policy. However, setting an excessively high intrinsic factor may lead to an overestimation bias, adversely affecting policy performance, as observed with $h = 0.8$ in Figure 8. Conversely, decreasing the intrinsic factor can enhance policy performance for tasks that are less sensitive to safety. But if the intrinsic factor is too small, it may fail to sufficiently counteract the underestimation bias, resulting in partial constraint violations, as demonstrated with $h = 0.4$ in Figure 8. In our experiments, $h$ is set to 0.6, which balance policy performance and constraint satisfaction across various tasks.

Furthermore, we provide a theoretical analysis of the intrinsic factor within the MICE framework. The Worst-Case Constraint Violation bound in CPO Achiam et al. (2017) is:

$$J_C(\pi_{k+1}) \leq d + \frac{\sqrt{2\delta}\gamma\epsilon_C^{\pi_{k+1}}}{(1-\gamma)^2} \tag{70}$$

In our paper, the bound is:

$$J_C(\pi_{k+1}) \leq d - I + \frac{\sqrt{2\delta}\gamma\epsilon_C^{\pi_{k+1}}}{(1-\gamma)^2} \tag{71}$$

where $\epsilon_C^{\pi_{k+1}} := \max_s |\mathbb{E}_{a\sim\pi_{k+1}}[A_C(s,a)]|$, $I = \mathbb{E}_{\tau|\pi_{k+1}}\left[\sum_{t=0}^{\infty} \gamma^t c_t^I\right]$. $c_t^I = \frac{h\gamma_I^k}{(1+e^{l_2(t)})}$,

where $l_2(t) = \sum_{i=0}^{n-1} \|W(\tau_i^m(t) - \tau(t))\|_2$. As $l_2(t) \geq 0$ and $0 < \gamma_I < 1$. Denote the minimum value of $c_t^I$ as $c_{min}^I$, then we get $0 \leq c_{min}^I \leq c_t^I \leq \frac{h}{2}$.

The term of $I$ in our theorem 2 is bounded by:

$$0 \leq \frac{c_{min}^I(1-\gamma^t)}{1-\gamma} \leq I \leq \frac{h(1-\gamma^t)}{2(1-\gamma)} \tag{72}$$

Then the upper bound $d - I + \frac{\sqrt{2\delta}\gamma\epsilon_C^{\pi_{k+1}}}{(1-\gamma)^2}$ in our paper satisfies:

$$d - \frac{h(1-\gamma^t)}{2(1-\gamma)} + \frac{\sqrt{2\delta}\gamma\epsilon_C^{\pi_{k+1}}}{(1-\gamma)^2} \leq d - I + \frac{\sqrt{2\delta}\gamma\epsilon_C^{\pi_{k+1}}}{(1-\gamma)^2} \leq d - \frac{c_{min}^I(1-\gamma^t)}{1-\gamma} + \frac{\sqrt{2\delta}\gamma\epsilon_C^{\pi_{k+1}}}{(1-\gamma)^2} \leq d + \frac{\sqrt{2\delta}\gamma\epsilon_C^{\pi_{k+1}}}{(1-\gamma)^2} \tag{73}$$

The right-hand side of the above equation is the upper bound in CPO, which indicates that our upper bound is smaller than that of CPO. We can adjust the scale of the left-hand side of the above equation by controlling the intrinsic factor $h$, which is characterized as risk preference. By increasing $h$, we obtain a safer but more conservative policy. Conversely, by decreasing $h$, we can get a high-performance policy with partial constraint violation. The results supporting this observation are illustrated in Figure 8.

### C.2.2 MEMORY CAPACITY

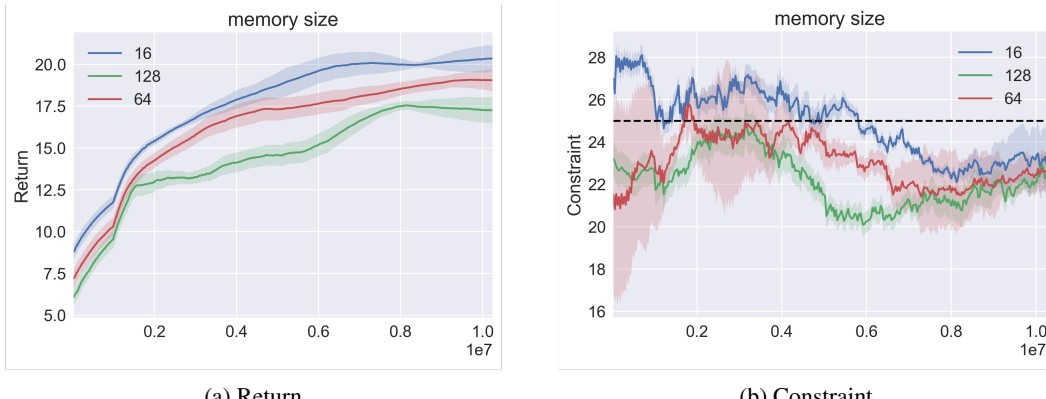

(a) Return        (b) Constraint

Figure 9: Comparative experiment on the effect of different **flashbulb memory capacity** on the performance of MICE-CPO algorithm in SafetyCarCircle1-v0 environment.

We conduct additional experiments in the SafetyCarCircle1-v0 environment, varying the effect of the flashbulb memory capacity in the MICE-CPO method. We experimented with memory capacity of

16, 64, and 128, with 64 being the value used in our original paper. The results of these experiments are shown in Figure 9.

With a memory capacity of 16, the results exhibit a more aggressive policy with more constraint violations. This occurs because a smaller memory can only store a limited number of unsafe trajectories. When the policy explores new danger zones, the memory may not provide sufficient intrinsic cost signals, leading to partial constraint violations but potentially higher performance.

Conversely, with a larger memory capacity of 128, the results show a more conservative policy with fewer constraint violations. A larger memory can store more past experiences, helping the agent avoid a greater number of known risks. However, for on-policy approaches, an excessively large memory may retain trajectories that are not relevant to the current policy. This misalignment can result in an overly conservative policy that hinders performance.

The performance is well-balanced at a medium memory capacity of 64, which was chosen in our original experiments. This capacity allows the agent to remember a sufficient number of past unsafe experiences, providing an adequate intrinsic cost signal to avoid known risks while maintaining robust policy performance.

### C.2.3 BASELINES

The MICE approach can be extended to other CRL algorithms based on actor-critic architectures. We compare MICE-CPO and MICE-PIDLag with their corresponding baselines, CPO and PIDLag, across multiple environments to validate the improvements offered by our approach. The results are shown in Figure 10 and Figure 11, which indicates that our MICE approach effectively improves constraint satisfaction over the respective original approaches while maintaining the same or even better level of policy performance.

We introduce more baselines to compare the effect of MICE. CUP Yang et al. (2022) is a projection approach that provides generalized theoretical guarantees for surrogate functions with a generalized advantage estimator Schulman et al. (2015), effectively reducing variance while maintaining acceptable bias. IPO Liu et al. (2020) augments the objective with a logarithmic barrier function to restrict the policy to feasible regions. P3O Zhang et al. (2022) penalizes constraints with a ReLU operator to obtain an unconstrained problem. We designed comparison experiments that include these baselines, as shown in Figure 12 and 13. These baselines are implemented based on the unified framework for safe RL in Ji et al. (2023). The results indicate that our approach outperforms baselines in both policy performance and constraint satisfaction across multiple tasks.

Here, we provide a additional introduction to the baseline methods employed in the main text. State augmentation methods aim to achieve constraint satisfaction with probability one. Saute RL Sootla et al. (2022a) eliminates safety constraints by expanding them into the state space and reshaping the objective. Specifically, the residual safety budget is treated as a new state to quantify the risk of violating the constraint. Simmer Sootla et al. (2022b) extends the state space with a state encapsulating the safety information. This safe state is initialized with a safety budget, and the value of the safe state can be used as a distance measure to the unsafe region. Simmer reduces safety constraint violations by scheduling the initial safety budget.

WCSAC Yang et al. (2021) is a constrained RL algorithm that extends the Soft Actor-Critic algorithm with a safety critic algorithm for risk control. It obtains a certain level of conditional Value-at-Risk (CVaR) from the distribution as a safety measure to judge constraint satisfaction. We construct experiments to compare WCSAC and MICE in the SafetyPointGoal1-v0 environment, with hyperparameters in WCSAC set as specified in its original paper. The results, shown in Figure 14, indicate that our method and WCSAC achieve similar results in terms of constraint satisfaction, MICE converges faster, as shown in Figure 14(b). In terms of cumulative rewards, MICE significantly outperforms WCSAC, as shown in Figure 14(a). The CVaR method in WCSAC focuses on the tail distribution of the cost value function, which can be affected by extreme data, resulting in overly conservative policy performance. In contrast, MICE effectively balances policy performance and constraint satisfaction by mitigating underestimation.

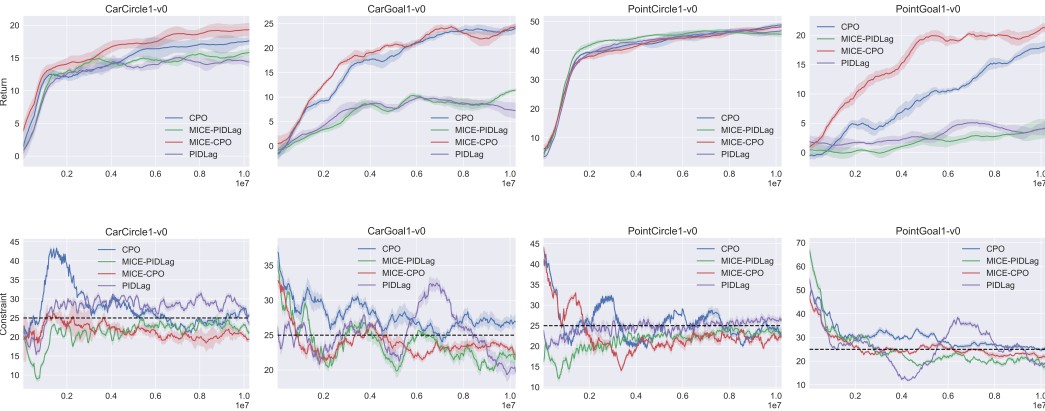

Figure 10: Comparison of MICE and their respective baseline approaches on Safety Gym. The x-axis is the total number of training steps, the y-axis is the average return or constraint. The solid line is the mean and the shaded area is the standard deviation. The dashed line in the cost plot is the constraint threshold which is 25.

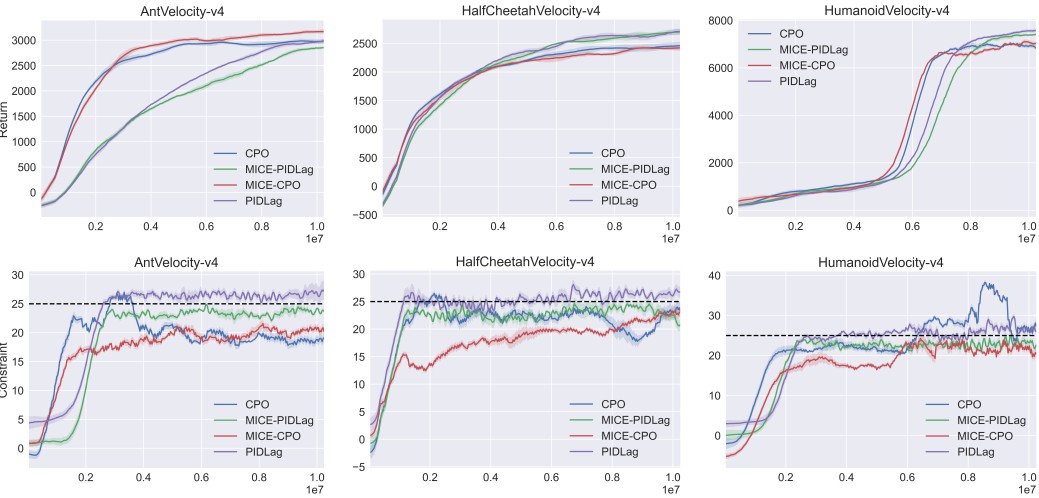

Figure 11: Comparison of MICE and their respective baseline approaches on Safety MuJoCo.

### C.2.4 COMPARED TO TD3-BASED COST VALUE FUNCTION

TD3 is a reinforcement learning method designed to mitigate the overestimation bias in the reward value function, which uses the minimum output from two separately-learned action-value networks during policy update. Similarly, TD3 can serve as a baseline for addressing underestimation bias in cost by using the maximum output from two separately-learned cost value networks. We conducted experiments to compare the cost value estimation bias between TD3 cost value function and MICE with the PIDLag optimization method in SafetyPointGoal1-v0 and SafetyCarGoal1-v0, as shown in Figure 15.

The results show that TD3 mitigates underestimation bias in cost value estimation, but it cannot fully eliminate it. This limitation arises from the inherent slow adaptation of neural networks, which results in a residual correlation between the value networks, thus preventing TD3 from completely eliminating the underestimation bias. In contrast, MICE can completely eliminate this bias by adjusting the intrinsic factor, leading to improved constraint satisfaction.

Additionally, compared to the TD3 cost value function, the flashbulb memory structures in MICE help address the catastrophic forgetting issue in neural networks Lipton et al. (2016), where agents may forget previously encountered states and revisit them under new policies. This mechanism

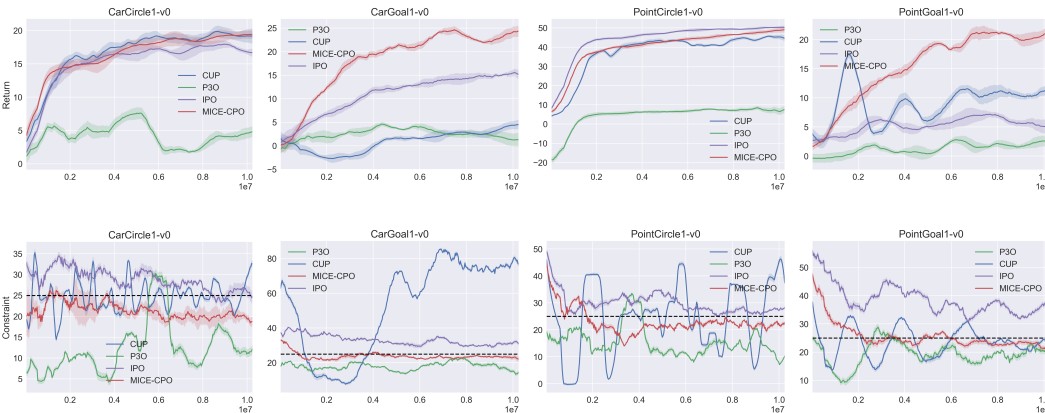

Figure 12: Comparison of MICE to baselines on Safety Gym. The x-axis is the total number of training steps, the y-axis is the average return or constraint. The solid line is the mean and the shaded area is the standard deviation. The dashed line in the cost plot is the constraint threshold which is 25.

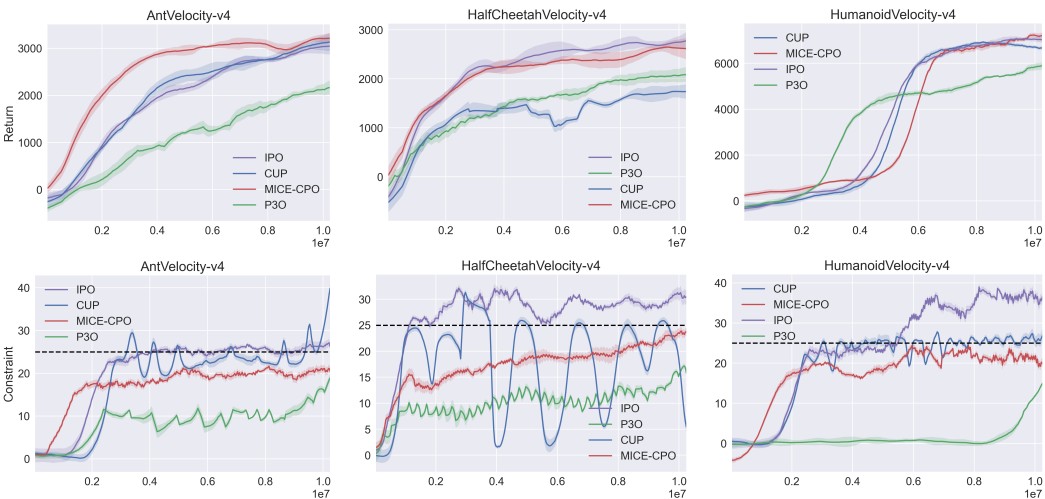

Figure 13: Comparison of MICE to baselines on Safety MuJoCo.

generates intrinsic cost signals that guide the agent away from previously explored dangerous trajectories, effectively preventing repeated encounters with the same hazards.

### C.2.5 EVALUATION ON MORE ENVIRONMENTS

We extended our evaluation to encompass more tasks and complex settings, including SafetyPointButton1-v0 and SafetyHopperVelocity-v4, with results presented in Appendix C.2.5 and Figure 16 (highlighted in green for clarity). The SafetyPointButton1-v0 task has more complex settings, with an observation dimensionality of 76 (range $(-\infty, \infty)$) and an action dimensionality of 2 (range $(-\infty, \infty)$). SafetyHopperVelocity-v4 introduces a different robot type. These experiment results demonstrate that MICE effectively balances performance and constraint satisfaction compared to multiple baselines, showcasing its scalability across a broader range of task types and robot types.

### C.2.6 CRITERIA FOR JUDGING UNSAFE TRAJECTORIES.

For risk-sensitive tasks, when many trajectories have costs just below the threshold, more conservative criteria can be adopted for judging unsafe trajectories. For instance, a trajectory can be stored in memory if its cumulative cost exceeds a specific quantile of the constraint threshold, thereby improving constraint satisfaction.

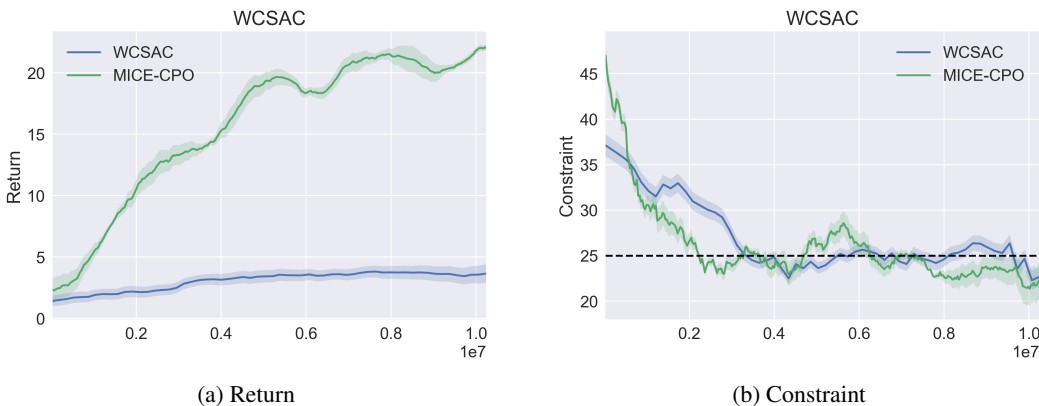

(a) Return              (b) Constraint

Figure 14: **(a)(b)** Comparison experiment between **MICE and WCSAC** in SafetyPointGoal1-v0.

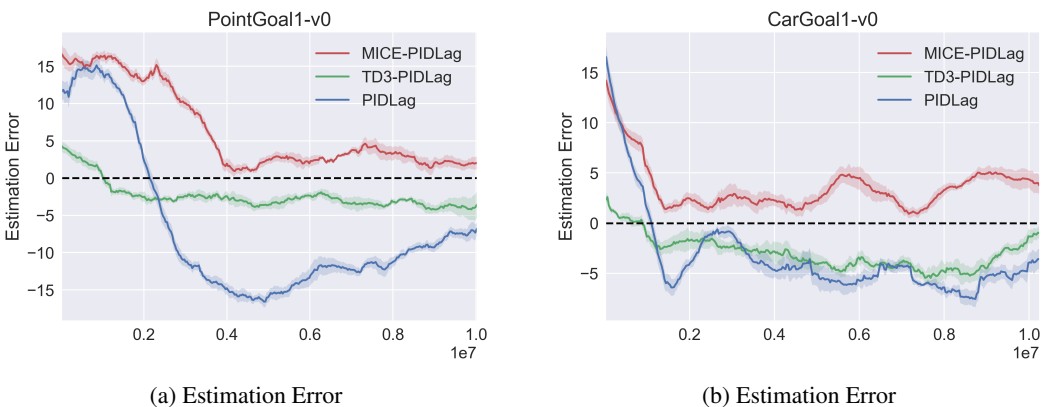

(a) Estimation Error              (b) Estimation Error

Figure 15: Comparison experiment about Estimation Error of MICE-PIDLag to TD3-PIDLag. The y-axis is the cost value estimate minus the true value, and the dashed line is the zero deviation.

We conducted experiments where trajectories were stored if their cumulative cost exceeded 90% of the constraint threshold. The results, as shown in Figure 17, demonstrate that this approach leads to a more conservative policy with reduced constraint violations.

## C.3 ENVIRONMENTS

### C.3.1 SAFETY GYM

Figure 18 shows the environments in the Safety Gym. Safety Gym is the standard API for safe reinforcement learning developed by Open AI. The agent perceives the world through the sensors of the robots and interacts with the environment via its actuators in Safety Gym. In this work, we consider two agents, Point and Car, and two tasks, Goal and Circle.

The Point is a simple robot constrained to a two-dimensional plane. It is equipped with two actuators, one for rotation and another for forward/backward movement. It has a small square in front of it, making it easier to visually determine the orientation of the robot. The action space in Point consists of two dimensions ranging from -1 to 1, and the observation space consists of twelve dimensions ranging from negative infinity to positive infinity.

The Car is a more complex robot that moves in three-dimensional space and has two independently driven parallel wheels and a freely rotating rear wheel. For this robot, both steering and forward/backward movement require coordination between the two drive wheels. The action space of Car includes two dimensions with a range from -1 to 1, while the observation space consists of 24 dimensions with a range from negative infinity to positive infinity.

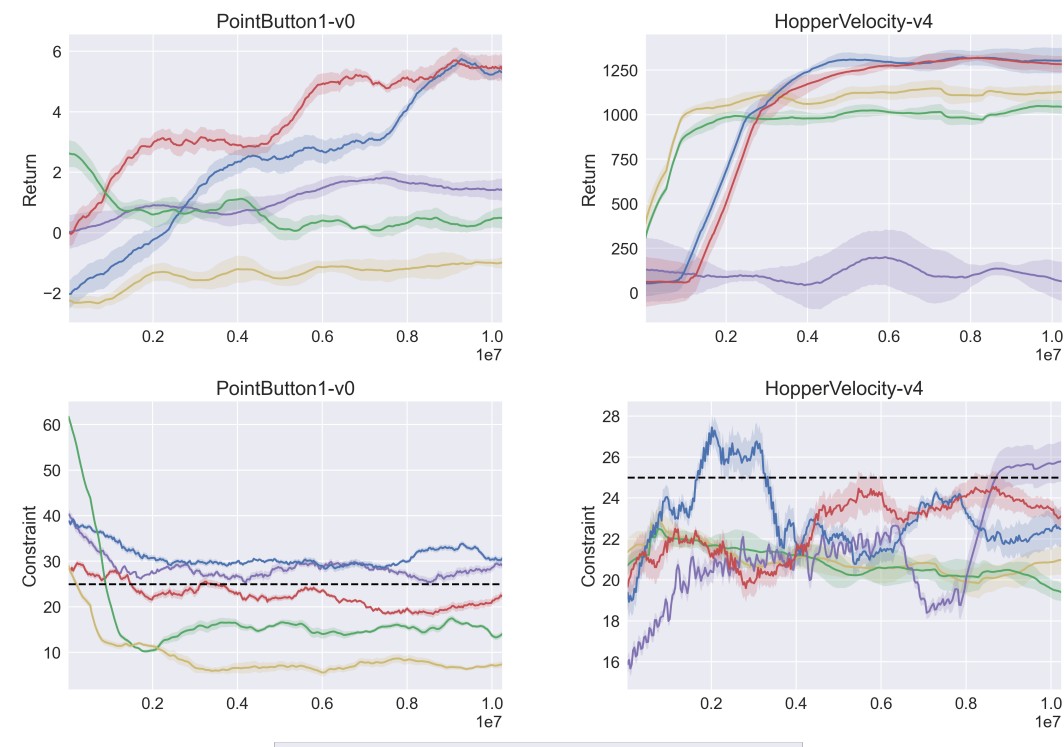

Figure 16: Comparison of MICE to baselines on More Environments. The x-axis is the total number of training steps, the y-axis is the average return or constraint. The solid line is the mean and the shaded area is the standard deviation. The dashed line is the constraint threshold which is 25.

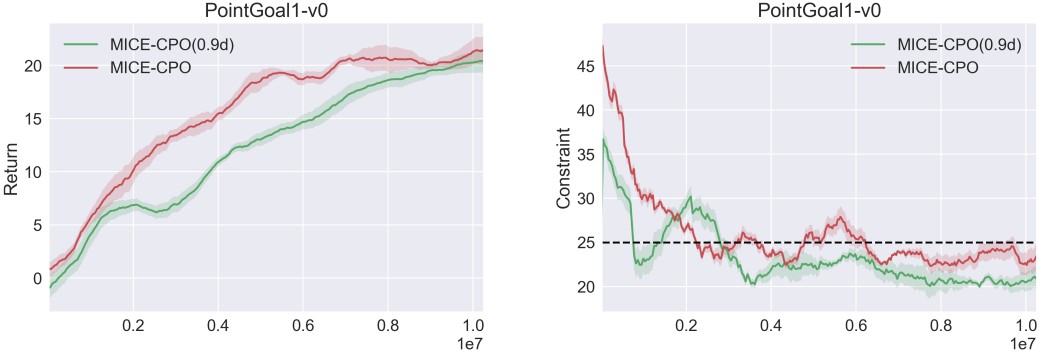

Figure 17: Comparison of MICE-CPO with different criteria for judging unsafe trajectories. MICE-CPO with cumulative cost exceeds constraint threshold, MICE-CPO(0.9d) with cumulative cost exceeds 90% of constraint threshold.

**Goal:** The agent is required to navigate towards the location of the goal. Upon successfully reaching the goal, the goal location is randomly reset to a new position while maintaining the remaining layout unchanged. The rewards in the task of Goal are composed of two components: reward distance and reward goal. In terms of reward distance, when the agent is closer to the Goal it gets a positive value of reward, and getting farther will cause a negative reward. Regarding the reward goal, each time the agent successfully reaches the Goal, it receives a positive reward value denoting the completion of the goal. In SafetyGoal1, the Agent needs to navigate to the Goal's location while circumventing Hazards. The environment consists of 8 Hazards positioned throughout the scene randomly.

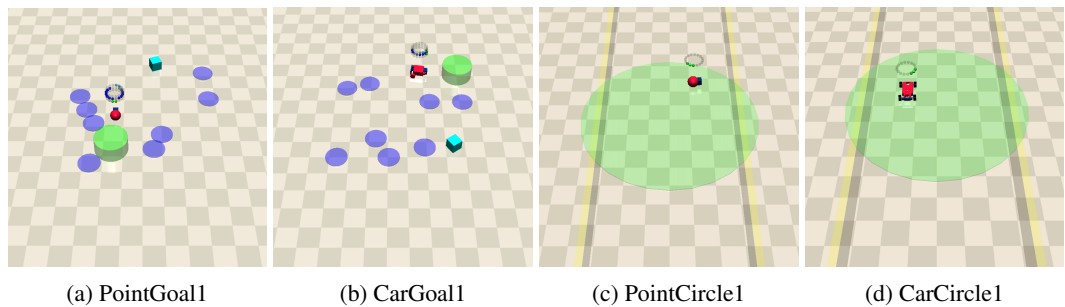

(a) PointGoal1      (b) CarGoal1      (c) PointCircle1      (d) CarCircle1

Figure 18: Environments in Safety Gym.

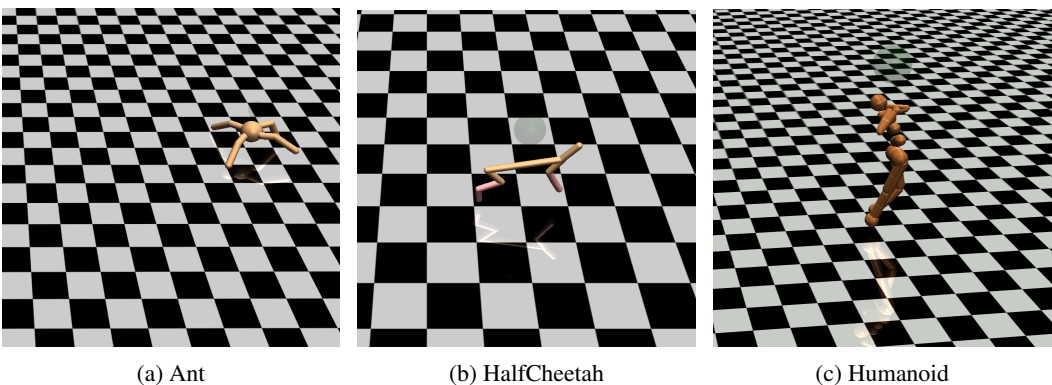

(a) Ant      (b) HalfCheetah      (c) Humanoid

Figure 19: Environments in Safety MuJoCo.

**Circle:** Agent is required to navigate around the center of the circle area while avoiding going outside the boundaries. The optimal path is along the outermost circumference of the circle, where the agent can maximize its speed. The faster the agent travels, the higher the reward it accumulates. The episode automatically ends if the duration exceeds 500 time steps. When out of the boundary, the agent gets an activated cost.

### C.3.2 SAFETY MUJOCO

The agent in Safety MuJoCo is provided by OpenAI Gym, and it is trained to move along a straight line while constrained with a velocity limit. Figure 19 illustrates the different environments.

### C.4 HYPERPARAMETERS

All experiments are implemented in Pytorch 2.0.0 and CUDA 11.3 and performed on Ubuntu 20.04.2 LTS with a single GPU (GeForce RTX 3090). The hyperparameters are summarized in Table 1.

| Parameter | CPO | PIDLag | MICE-CPO | MICE-PIDLag |
|---|---|---|---|---|
| hidden layers | 2 | 2 | 2 | 2 |
| hidden sizes | 64 | 64 | 64 | 64 |
| activation | $tanh$ | $tanh$ | $tanh$ | $tanh$ |
| actor learning rate | $3e-4$ | $3e-4$ | $3e-4$ | $3e-4$ |
| critic learning rate | $3e-4$ | $3e-4$ | $3e-4$ | $3e-4$ |
| intrinsic weight $\omega$ | $N/A$ | $N/A$ | 0.5 | 0.5 |
| batch size | 64 | 64 | 64 | 64 |
| trust region bound | $1e-2$ | $N/A$ | $1e-2$ | $N/A$ |
| discount factor gamma | 0.99 | 0.99 | 0.99 | 0.99 |
| GAE gamma | 0.95 | 0.95 | 0.95 | 0.95 |
| intrinsic discount factor gamma $\gamma_I$ | $N/A$ | $N/A$ | 0.99 | 0.99 |
| normalization coefficient | $1e-3$ | $1e-3$ | $1e-3$ | $1e-3$ |
| clip ratio | $N/A$ | 0.2 | $N/A$ | 0.2 |
| conjugate gradient damping | 0.1 | $N/A$ | 0.1 | $N/A$ |
| initial lagrangian multiplier | $N/A$ | $1e-3$ | $N/A$ | $1e-3$ |
| lambda learning rate | $N/A$ | 0.035 | $N/A$ | 0.035 |
| intrinsic factor $h$ | $N/A$ | $N/A$ | 0.6 | 0.6 |
| memory capacity $n$ | $N/A$ | $N/A$ | 64 | 64 |

Table 1: Hyperparameters

