# OpenReview forum: "MICE: Memory-driven Intrinsic Cost Estimation for Mitigating Constraint Violations"
_ICLR.cc/2025/Conference — Submitted to ICLR 2025_

### Official Review · Reviewer_xyec · 2024-11-03

**Soundness:** 3
**Presentation:** 2
**Contribution:** 3
**Rating:** 6
**Confidence:** 3

**Summary:**

The authors present MICE, a method to combat the cost underestimation problem in constrained RL. CRL cost value functions often underestimate the cost of actions taken, resulting in risky actions being taken more often than they should.

In order to address this, the authors propose a few key innovations:
1. An explicit memory storage system for bad trajectories that violate some threshold of cost.
2.  A generator model that determines an 'intrinsic' cost value for trajectories based on the stored dangerous trajectories
3. The intrinsic cost is added to the extrinsic cost, training the cost critic to estimate bad state-action pairs closer to their true cost.

**Strengths:**

This work is well-motivated in that it addresses an as-of-now open and broad problem in CRL, which is as much a question of study as overestimation is in value-function RL. Their idea to mitigate underestimation by explicitly storing violating trajectories seems sound as it helps counter some potential problems that could arise with constrained cost estimation in RL using function approximation such as loss of network plasticity or catastrophic forgetting.

The bound provided in Theorem 1 also lends credibility to the idea, as it shows (seemingly convincingly based on the proof provided in the appendix) that this method should guarantee an improvement in cost estimation.

The results in Figures 3 and 4 indicate robustness to the performance criteria while maintaining constraint validity across the environments tested. While MICE does not always reach the performance of competing baselines, it does so by ensuring coherence with the constraint thresholds indicated, which is the desirable outcome.

The ablations also provide interesting insight into the effects of changing the key hyperparameters such as constraint threshold and memory capacity, where the performance/constraint behaviour of the method changes as expected, which indicates that the intuition behind these mechanisms inferred by the authors is correct.

The clarity of the main text is fairly high and consistent across the paper, and the authors avoid bloviation or confusion by drawing clear conclusions and analogies (e.g. the comparison to the bio-inspired traumatic memory mechanism in humans is intuitive and sets up the main idea nicely).

**Weaknesses:**

The method requires setting hyperparameters which have non-trivial effects on the performance and constraint following of the method, resulting in a potential complication in practice. How were these hyperparameters set in the paper? How can they be appropriately determined by a practitioner hoping to use this method for a problem where, for e.g. the acceptable level of constraint violation is subjective?

The use of explicit memory places an arbitrary distinction between acceptable and unacceptable trajectories, which will affect how the intrinsic costs are determined. If adhering to an arbitrary constraint level is required, this may be appropriate. However, what if many trajectories have costs just below the threshold and are therefore not stored?

Results:
- It is not clear why the standard deviation of results was provided, when the 95% confidence interval would probably be a more appropriate measure of uncertainty across runs.

Presentation:
- The plots are not always the clearest, as axes are sometimes labelled and sometimes not
- The plot legends do not indicate what value is changing (this has to be inferred from the plot titles and the main text)

**Questions:**

It is not entirely clear to me why a generator network is required as it would appear that an explicit memory and trajectory-distance metric are enough for the rest of the method to work. Why not simply store violating trajectories and then provide an intrinsic cost based on new trajectories' distance to the stored trajectories? What benefit does the generator provide?

---

> ### Author Response · Authors · 2024-11-23
>
> We sincerely appreciate the reviewer's insightful comments. The following are the detailed responses to your comments.
>
>
> **Weakness 1**
> > How were these hyperparameters set in the paper? How can they be appropriately determined by a practitioner hoping to use this method for a problem where, for e.g. the acceptable level of constraint violation is subjective?
>
> **Response:** We appreciate the reviewer's valuable comment and provide a detailed explanation of the hyperparameters.
> The intrinsic factor $h$ is a critical hyperparameter in MICE, set to 0.6 in this paper.
> By adjusting $h$, we can control the acceptable level of constraint violation to meet different task requirements, as shown in Figure 7(a)(b). For tasks prioritizing safety, increasing $h$ enhances risk aversion and improves constraint satisfaction. Conversely, for tasks emphasizing performance, decreasing $h$ allows for better policy performance.
>
> Memory capacity, set to 64 in this paper, has a comparatively smaller impact on MICE's performance, as shown in Figures 7(c) and 7(d).
> We provide a detailed analysis of these hyperparameters in Appendix C.2.1 and C.2.2.
>
>
> **Weakness 2**
> > What if many trajectories have costs just below the threshold and are therefore not stored?
>
> **Response:** Thank you for your valuable comment. For risk-sensitive tasks, when many trajectories have costs just below the threshold, more conservative criteria can be adopted for judging unsafe trajectories. For instance, a trajectory can be stored in memory if its cumulative cost exceeds a specific quantile of the constraint threshold, thereby improving constraint satisfaction.
>
> We have conducted experiments where trajectories were stored if their cumulative cost exceeded 90\% of the constraint threshold. The results, detailed in the Appendix C.2.6 and Figure 17 (highlighted in red for clarity), demonstrate that this approach leads to a more conservative policy with reduced constraint violations.
>
>
> **Weakness 3**
> > Why the standard deviation of results was provided?
>
> **Response:** We appreciate the reviewer's careful review and valuable comments. To ensure fairness in the experimental results, we adopt the same evaluation method using the standard deviation as employed in the baseline works.
>
>
> **Weakness 4**
> > The plots are not always the clearest, as axes are sometimes labelled and sometimes not. The plot legends do not indicate what value is changing.
>
> **Response:** Thank you for your valuable comment. We have modified the revision according to your suggestions to improve clarity.
>
>
> **Question 1**
> > Why a generator network is required as it would appear that an explicit memory and trajectory-distance metric are enough for the rest of the method to work. Why not simply store violating trajectories and then provide an intrinsic cost based on new trajectories' distance to the stored trajectories? What benefit does the generator provide?
>
> **Response:** We appreciate the reviewer's insightful questions and have provided a detailed explanation about this in the revision for clarity.
>  The Generator is designed to minimize memory accesses, thereby reducing computational complexity. During the agent’s learning process, if the cumulative cost of the current trajectory $\tau$ is below the constraint threshold, the generator directly produces the intrinsic cost.
> Additionally, the generator network’s representation capability enhances the generalization of intrinsic cost signals.
>
> We hope our responses address your comments. Please let us know if you have follow-up questions. We would be happy to discuss.

---

> > ### Comment · Reviewer_xyec · 2024-12-02
> >
> > Thank you for the clarifications, the results of the work are more easily understandable now.
> >
> > As it stands, I believe the motivation for using the generator is somewhat convincing. While it may reduce memory accesses, a generator still needs to be re-trained when new violating trajectories are found, which certainly does a computational overhead to this method. Some analysis of total run time or compute needed would help motivate this aspect of the work, as well as a study of the generator's ability to generalise well to new trajectories.
> >
> > The general idea of the work, i.e. using some notion of proximity to violating trajectories to generate an intrinsic cost, is its strongest aspect. For this reason, I believe my current score is accurate.

---

> > > ### Author Response · Authors · 2024-12-02
> > >
> > > We sincerely appreciate you reviewing our response and maintaining a positive stance.
> > >
> > > Based on your insightful comment, we have provided the average runtime of MICE-CPO and the baselines in the SafetyPoinCircle1-v0 environment. All experiments were implemented on a PC with a single GPU (GeForce RTX 3090).
> > > The results, as shown in the table below, indicate that MICE-CPO does result in a slight increase of 7.49% in runtime compared to CPO without flashbulb memory. The generator design in MICE minimizes memory access, and constraint violations are rare in the later stages of training, leading to a reduced frequency of generator updates. As a result, the increased training time remains within an acceptable range.
> > >
> > > We acknowledge that, compared to existing methods, the MICE approach does not offer improvements in computational speed. However, the introduction of flashbulb memory in MICE has led to improvements in constraint satisfaction and policy performance. These benefits are demonstrated across multiple tasks.
> > >
> > >
> > >
> > >
> > > |      | MICE-CPO(ours) |  CPO   |    PIDLag   |   Saute |   SimmerPID   |    CUP    |
> > > | :---        |   :----:   |   :----:   |   :----:   |  :----:  |  :----:   |   ----:   |
> > > | Time      |    16h 36m 19s    |    15h 26m 51s   |   15h 13m 5s   |   16h 43m 57s    |   16h 56m 23s    |  17h 19m 39s   |
> > >
> > > Due to current upload limitations, we will include the corresponding experimental results and analysis in the next revision.

---

### Official Review · Reviewer_c9zg · 2024-11-03

**Soundness:** 2
**Presentation:** 2
**Contribution:** 2
**Rating:** 3
**Confidence:** 4

**Summary:**

This paper proposes the MICE algorithm, designed to address constraint violations in constrained reinforcement learning by mitigating the underestimation of cost values. MICE introduces an intrinsic cost signal inspired by flashbulb memory, which stores and references unsafe trajectories to adjust cost estimates for actions likely to result in constraint violations. Empirical results on various safety-critical environments demonstrate a reduction in constraint violations with baselines.

**Strengths:**

MICE introduces a flashbulb memory module that stores unsafe trajectories, which is a novel approach in CRL to handle underestimation biases and improve constraint adherence.
Clear Problem Identification: The paper identifies and targets a critical issue in CRL: the underestimation of cost values leading to unsafe policy behavior, making this contribution relevant for safety-critical applications.
Theoretical and Empirical Contributions: MICE provides a theoretical framework with guarantees on constraint violation and convergence, supported by empirical results across various environments, reinforcing the algorithm’s applicability.

**Weaknesses:**

- The writing of this paper requires significant improvement and polish. Many unclear notations and statements make it hard to follow. For example, in equations (3) and (4), the symbols $\tau_i^m(t)$ and $\tau(t)$ are never defined. The term $c_t^I$ is stated to be a function of $\gamma_I^k$, where $k$ represents the iteration number; thus, $c_t^I$ should also depend on $k$. The symbol $W$ is described as a set, yet in equation (3), $W$ appears to be used as a weight. Furthermore, it is unclear what $n$ represents in this context. Additionally, $c^E$ is not defined in equation (4).

- In the definition of $J_C^{EI}$, which depends on both extrinsic and intrinsic costs, $c^E$ seems to rely on flashbulb memory. This makes it a random variable dependent on the policy applied at each step. Fundamentally, the expectations in $J_C^{EI}$ and $J_C(\pi)$ differ, and thus, one cannot prove lemmas such as Lemma 1 by simply subtracting one from the other.

- I observed no discussion on the reward function. Typically, in safe RL, actions are chosen not solely based on the $Q$ function. Therefore, the Bellman equation presented in (6) does not appear to be correct. As mentioned in the introduction, safe RL is often solved using a primal-dual approach, which implies that the value function update should be in the SARSA form.

- Based on the experimental results, it seems the paper’s claimed contribution towards addressing training violations is not clearly substantiated.

**Questions:**

1. Over what is the expectation in the equation (5) taken?

2. Are there any justifications for calculating the difference in equation (3) in terms of individual steps? Two safe trajectories may not even contain the same state-action pairs at each step.

3. LP-based approaches are missing in the related work.

4. Typo in Figure 6? The word "constant" appears on top of each figure.

---

> ### Author Response · Authors · 2024-11-23
>
> We sincerely appreciate the reviewer's insightful comments. The following are the detailed response for your comments.
>
>
> **Weakness 1**
> > The writing of this paper requires significant improvement and polish. Many unclear notations and statements make it hard to follow. For example, in equations (3) and (4), the symbols $\tau_i^m(t)$ and $\tau(t)$ are never defined. The term $c^I_t$ is stated to be a function of $\gamma_I^k$, where $k$ represents the iteration number; thus, $c_t^I$ should also depend on $k$. The symbol $W$ is described as a set, yet in equation (3), $W$ appears to be used as a weight. Furthermore, it is unclear what $n$ represents in this context. Additionally, $c^E$ is not defined in equation (4).
>
> **Response:** Thank you for your valuable comments. We have made modifications to this part in the revision to improve clarity and readability (highlighted in red).
> $\tau(t)=\{s_0, a_0, \cdots, s_{t-1}, a_{t-1}\}$ denotes the current trajectory from the initial time step to $t$.
> $\tau_i^m(t)=\tau_i^m[0:t-1] $ denotes the segment of the $i$-th unsafe trajectory in memory $M$ from the initial time step to $t$.
> $W$ denotes the weight vector, and $n$ denotes the number of unsafe trajectories stored in memory. $c^E$ denotes the extrinsic cost, which is provided by the environment.
> We also provided the Notations Table in the first page of the Appendix.
>
>
> **Weakness 2**
> > In the definition of $J_C^{EI}$, which depends on both extrinsic and intrinsic costs, $c^E$ seems to rely on flashbulb memory. This makes it a random variable dependent on the policy applied at each step. Fundamentally, the expectations in $J_C^{EI}$ and $J_C(\pi)$ differ, and thus, one cannot prove lemmas such as Lemma 1 by simply subtracting one from the other.
>
> **Response:** We appreciate the reviewer's careful observations. We would like to provide a detailed explanation of Lemma 1 and its proof.
> $c^E$ denotes the extrinsic cost, which is task-related and generated by the environment, independent of the flashbulb memory. $c^I$ denotes the intrinsic cost generated by the flashbulb memory. Both $c^E(s_t,\pi(s_t))$ and $c^I(s_t,\pi(s_t))$ depend on the policy $\pi$.
>
> The term $E_{\tau|\pi'} \left[ \sum_{t=0}^\infty \gamma^t A_{C}^{EI}(s_t,a_t|\pi) \right]$ in Lemma 1 indicates that using the advantage function $A^{EI}_C$ over $\pi$ to evaluate the trajectory generated by $\tau \sim \pi'$, similar to the performance difference lemma in TRPO.
>
> The expectations in $J_C^{EI}(\pi')$ and $J_C(\pi)$ can be expanded as:
> \begin{equation}
>  J_C^{EI}(\pi'):= E_{\tau \sim \pi'} [\sum_{t=0}^{\infty} \gamma^t (c^E(s_t, \pi'(s_t)) + c^I(s_t, \pi'(s_t)))]
> \end{equation}
> \begin{equation}
>     J_C(\pi):= E_{\tau \sim \pi} [\sum_{t=0}^{\infty} \gamma^t c^E(s_t,\pi(s_t))] = \mathbb{E}_{s_0\sim\rho}[V_C^\pi(s_0)]
> \end{equation}
>
> The trajectories generated by $\pi'$ remain unchanged during the proof. The expectation of  $J_C(\pi)$ can be denoted as $\mathbb{E}_{s_0\sim\rho}[V_C^\pi(s_0)]$, which independent of the trajectories $\tau \sim \pi'$.
> The initial state $s_0$ in $V_C^\pi(s_0)$ depends solely on the initial state distribution $\rho$, allowing the expectation over $s_0 \sim \rho$ to be expressed as an expectation over $\tau|\pi'$.
> Therefore, the expectations in $J_C^{EI}$ and $J_C(\pi)$ can be subtracted.
>
> Due to word limitations, we provide a detailed explanation of the proof in the Appendix B.2.

---

> ### Author Response · Authors · 2024-11-23
>
> **Weakness 3**
> > The Bellman equation presented in (6) does not appear to be correct. As mentioned in the introduction, safe RL is often solved using a primal-dual approach, which implies that the value function update should be in the SARSA form.
>
> **Response:** We appreciate the reviewer's careful review and valuable comment. We have clarified this point in revision.
> Equation (6) applies to the Q-learning case, corresponding to the analysis in Section 4.1.
> In our work, both MICE-CPO and MICE-PPOLag are PPO-based methods, with their cost value updates employing the Generalized Advantage Estimation (GAE) method [1], as shown in Appendix C.1:
>
> $ \hat{A}^{EI}_t = \sum _ {l=0}^\infty (\gamma \lambda)^l \delta _ {t+l}$
>
> where $\delta_t = c^E_t + c^I_t + \gamma V(s_{t+1}) - V(s_t)$.
> The details of GAE can be found in [1].
>
>
> **Weakness 4**
> > Based on the experimental results, it seems the paper’s claimed contribution towards addressing training violations is not clearly substantiated.
>
> **Response:** Thank you for your valuable comment.
> By increasing the intrinsic factor, MICE can achieve zero constraint violations during the learning process, as demonstrated in Figures 7(a) and 7(b).
>
> By adjusting $h$, we can control the acceptable level of constraint violation to meet different task requirements. For tasks prioritizing safety, increasing $h$ enhances risk aversion and improves constraint satisfaction. Conversely, for tasks emphasizing performance, decreasing $h$ allows for better policy performance. The detailed analysis is provided in Appendix C.2.1.
>
>
>
> **Questions 1**
> > Over what is the expectation in the equation (5) taken?
>
> **Response:** Thank you for your valuable comment. We have clarified this point in revision.
> The expectation in equation (5) is taken over trajectories $\tau$ where the cumulative extrinsic cost exceeds the constraint threshold during the agent's learning process.
>
>
> **Questions 2**
> > Are there any justifications for calculating the difference in equation (3) in terms of individual steps?
>
> **Response:** Thank you for your insightful question. If the current trajectory shares more similar state-action pairs with an unsafe trajectory in flashbulb memory, it indicates a higher risk that the current trajectory similarly violates the constraint as the unsafe trajectory.
> The purpose of Equation (3) is to direct the current trajectory away from previously encountered unsafe trajectories and prioritize segments with significant costs.
>
>
> **Questions 3**
> > LP-based approaches are missing in the related work.
>
> **Response:** Thanks to your reminder, we have added the relevant method in the related work of the revision.
>
>
> **Questions 4**
> > Typo in Figure 6? The word "constant" appears on top of each figure.
>
> **Response:** Thank you for your careful comment, and we have made it more clear in the revision.
>
>
>
>
> We hope our responses address your comments. Please let us know if you have follow-up questions. We would be happy to discuss.
>
>
> **References**
> >[1] John Schulman, Philipp Moritz, Sergey Levine, Michael Jordan, and Pieter Abbeel. High-
> dimensional continuous control using generalized advantage estimation. arXiv preprint
> arXiv:1506.02438, 2015.

---

> > ### Author Response · Authors · 2024-12-02
> >
> > Dear Reviewer c9zg,
> >
> > I hope this message finds you well. I want to kindly follow up regarding our response to your comments. We highly value your feedback and have provided detailed answers to the weaknesses and questions you raised.
> >
> > It would mean a great deal to us to know whether we have adequately addressed your concerns. Please let us know if there is anything further we can clarify.
> >
> > Thank you again for your time and thoughtful review. We sincerely appreciate your efforts in helping us improve our work.
> >
> > Best regards.

---

### Official Review · Reviewer_JSW4 · 2024-11-04

**Soundness:** 2
**Presentation:** 3
**Contribution:** 2
**Rating:** 5
**Confidence:** 3

**Summary:**

This paper investigates underestimation bias of cost value function in actor-critic based safe RL algorithms. It provides an overview on why this underestimation occurs and proposes MICE method which learns an "intrinsic cost" to compensate this underestimation bias. The contributions include (i) designed flashbulb memory module which outputs intrinsic cost which is subsequently used to mitigate underestimation in cost-value function, (ii) provided theoretical bound on the worst-case constraint violation for MICE update and convergence guarantee, (iii) performed a number of empirical experiments to ascertain the validity of the proposed approach.

**Strengths:**

1. The paper is written in an easy-to-understand manner and addresses an important problem in safe RL: value underestimation, which impacts most (if not all) safe RL algorithms.

2. The motivation behind this paper is clear and it provided useful theoretical analyses demonstrating the convergence guarantee and bound on constraint violation.

3. The paper performed a variety of experiments showing the outperformance of its method, sensitivity analysis of hyper-parameters, ablation study and robustness validation.

**Weaknesses:**

1. Baseline which addresses underestimation bias

I understand that the underestimation bias was caused by the $min$ operator when safe RL algorithm is trying to fit the action-value function for cost $\hat{Q}_c$. This is analogous to the overestimation bias for reward when RL algorithm uses $max$ operator while fitting action-value function $\hat{Q}_r$.

For the overestimation bias, what RL algorithm (e.g. TD3) did was that it uses the minimum of the output from two separately-learned action-value networks $\hat{Q}_r^1, \hat{Q}_r^2$ for policy update. Similarly, can't we use the maximum of the output from two separately-learned action-value networks for cost $\hat{Q}_c^1, \hat{Q}_c^2$ to combat this underestimation bias in cost? This sounds like a possible simpler baseline to me.

2. Elaboration on Eq3

I think it'd be good for the paper to discuss how they design the formula for intrinsic cost. I can't fully grasp why the intrinsic cost is designed this way just by reading the main paper. For example:

a. What does the discount factor here signify? This "intrinsic discount factor" is different from the reward and cost discount factor and its exponent is $k$: iteration number of value estimation. Why is there compounding discount as the iteration increases?

b. The L2-norm distance depends on time index $t$ such that it only compares the state-action similarity (between current trajectory and historical trajectory) at time $t$. This is quite counterintuitive because RL policy function or value function are usually not time-index dependent.

c. The weight $\omega$ seems to be another hyper-parameter. How does one decide the right weight to use? The paper does state that $\omega$ should be between 0 and 1 but finding the right value still seems to be tricky.

3. Explanation on how MICE eliminates underestimation bias

The paper mentions that MICE injects overestimation into the cost value-function estimate. Injecting overestimation bias to an underestimated function does not mean the underestimation bias is correctly eliminated since the injected overestimation could under-compensate or over-compensate the pre-existing underestimation bias.

I think the paper could perhaps elaborate more on how the overestimation introduced by MICE correctly mitigates the underestimation bias.

4. Minor typo: "worst-case" in line 073

**Questions:**

Please refer to the weaknesses section as all my questions have been listed there. I'm more than happy to discuss and please let me know if I misunderstood or misses out anything.

---

> ### Author Response · Authors · 2024-11-23
>
> We sincerely appreciate the reviewer's insightful comments. The following are the detailed response for your comments.
>
> **Weakness 1**
> > A baseline like TD3 addresses underestimation bias by using the maximum of the output from two separately-learned action-value networks for cost to combat this underestimation bias in cost.
>
> **Response:**
> Thank you for your valuable comment. We have conducted experiments to compare the cost value estimation bias between TD3 cost value function and MICE with the PIDLag optimization method in SafetyPointGoal1-v0 and SafetyCarGoal1-v0, the results are shown in Appendix C.2.4 and Figure 15 (highlighted in red for clarity).
>
> The results show that TD3 mitigates underestimation bias in cost value estimation, but it cannot fully eliminate it. This limitation arises from the inherent slow adaptation of neural networks, which results in a residual correlation between the value networks, thus preventing TD3 from completely eliminating the underestimation bias. In contrast, MICE can completely eliminate this bias by adjusting the intrinsic factor, leading to improved constraint satisfaction.
>
> Additionally, compared to the TD3 cost value function, the flashbulb memory structure in MICE helps address the catastrophic forgetting issue in neural networks [1], where agents may forget previously encountered states and revisit them under new policies. The MICE mechanism generates intrinsic cost signals that guide the agent away from previously explored dangerous trajectories, effectively preventing repeated encounters with the same hazards.
>
>
> **Weakness 2**
> > Elaboration on Eq3
>
> **Response:** Thank you for your valuable comments. We have made modifications to this part in the revision to improve clarity and readability (highlighted in red).
>
> > a. What does the discount factor here signify? This "intrinsic discount factor" is different from the reward and cost discount factor and its exponent is $k$: iteration number of value estimation. Why is there compounding discount as the iteration increases?
>
> **Response:** The intrinsic discount factor $\gamma_I$ works with the iteration number $k$ of the value function update, which differs from the discount factor $\gamma$ used for rewards, which operates with the rollout time step $t$.
> The intrinsic discount factor $\gamma_I$ is designed to ensure that our cost value converges to the optimal value.
> As shown in traditional value function [2], increasing $k$ leads to a gradual convergence of the value estimate to the optimal value, with the underestimation bias decreasing. The term $\gamma^k$ can ensure that our proposed value function converges to the optimal value, as detailed in Theorem 3 with the convergence analysis.
>
>
> > b. The L2-norm distance depends on time index $t$ such that it only compares the state-action similarity (between current trajectory and historical trajectory) at time $t$. This is quite counterintuitive because RL policy function or value function are usually not time-index dependent.
>
> **Response:** In equation (3), $t$ represents the current time step of agent's rollout in environment. $\tau(t)= \{ s_0, a_0, \cdots, s_{t-1}, a_{t-1} \}$ denotes the current trajectory from the initial time step to $t$.
> $\tau_i^m(t)=\tau_i^m[0:t-1] $ denotes the segment of the $i$-th unsafe trajectory in memory $M$ from the initial time step to $t$.
> At each time step $t$, the flashbulb memory mechanism generates the intrinsic cost $c^I(s_t, \pi(s_t))$ as defined in equation (3), which corresponds to the extrinsic cost $c^E(s_t, \pi(s_t))$ provided by the environment. The cost is then used to update the value function $V_C(s_t)$ and policy $\pi(s_t)$ at the current time step $t$.
>
>
>
> > c. The weight $\omega$ seems to be another hyper-parameter. How does one decide the right weight to use?
>
> **Response:** When calculating the trajectory distance, $\omega$ is used to adjust the weights of different types of states to generate the intrinsic cost in MICE.
> We tested across multiple environments, demonstrating that $\omega = 0.5$ is a robust choice for achieving balanced performance and constraint satisfaction, as shown in Appendix C.4.
>
>
>
> **References**
> >[1] Zachary C Lipton, Kamyar Azizzadenesheli, Abhishek Kumar, Lihong Li, Jianfeng Gao, and Li Deng. Combating reinforcement learning’s sisyphean curse with intrinsic fear. arXiv preprint arXiv:1611.01211, 2016.
>
> >[2] Scott Fujimoto, Herke Hoof, and David Meger. Addressing function approximation error in actorcritic methods. In International conference on machine learning, pp. 1587–1596. PMLR, 2018.

---

> > ### Author Response · Authors · 2024-11-23
> >
> > **Weakness 3**
> > > How the overestimation introduced by MICE correctly mitigates the underestimation bias.
> >
> > **Response:** Thank you for your valuable suggestions.
> > Fig. 5(a)(b) shows that MICE introduces a partial overestimation bias compared to the true cost value. However, the overestimation bias is limited in propagation, as the policy tends to avoid actions with high-cost estimates. Overestimation in the extrinsic-intrinsic value function also helps correct estimation bias in high-value regions [3], as detailed in Appendix B.1.
> >
> > MICE can adjust the intrinsic factor to regulate underestimation bias based on task risk preferences, with a sensitivity analysis provided in Appendix C.2.1.
> > For low-risk preference, increasing the intrinsic factor fully eliminates underestimation bias, resulting in a more conservative cost value function that reduces constraint violations. For high-risk preference, decreasing the factor partially mitigates the bias, balancing policy performance with constraint satisfaction.
> >
> >
> >
> > **Weakness 4**
> > > Minor typo: "worst-case" in line 073.
> >
> > **Response:** Thank you for your careful review, we have corrected it in the revision.
> >
> >
> >
> > We hope our responses address your comments. Please let us know if you have follow-up questions. We would be happy to discuss.
> >
> >
> > **References**
> >
> > >[3] Thommen George Karimpanal, Hung Le, Majid Abdolshah, Santu Rana, Sunil Gupta, Truyen Tran, and Svetha Venkatesh. Balanced q-learning: Combining the influence of optimistic and pessimistic targets. Artificial Intelligence, 325:104021, 2023.

---

> > > ### Author Response · Authors · 2024-12-02
> > >
> > > Dear Reviewer JSW4,
> > >
> > > I hope this message finds you well. I want to kindly follow up regarding our response to your comments. We highly value your feedback and have provided detailed answers to the weaknesses and questions you raised.
> > >
> > > It would mean a great deal to us to know whether we have adequately addressed your concerns. Please let us know if there is anything further we can clarify.
> > >
> > > Thank you again for your time and thoughtful review. We sincerely appreciate your efforts in helping us improve our work.
> > >
> > > Best regards.

---

> > > > ### Comment · Reviewer_JSW4 · 2024-12-03
> > > >
> > > > I thank the authors for the detailed rebuttal response and the new experiments involving TD3-PIDLag in Fig15.
> > > >
> > > > May I check how do I read Fig15? In Fig15, the green curve (TD3-PIDLag) seems to have some underestimation bias (compared to ground truth) but the degree of underestimation is not too large. It seems to me that the absolute magnitude of the underestimation bias in green curve is comparable (if not better) than the overestimation bias in red curve.

---

> > > > > ### Author Response · Authors · 2024-12-03
> > > > >
> > > > > We sincerely thank you for reviewing our response.
> > > > >
> > > > > The y-axis in Figure 15 denotes the cost value estimate minus the true value, with the dashed line indicating zero deviation. The green curve (TD3-PIDLag) lies below the zero deviation line and above the blue curve (PIDLag), indicating that TD3 partially mitigates underestimation but cannot fully address it. In contrast, the red curve (MICE) lies above the zero deviation line, showing that MICE completely eliminates underestimation with introducing some overestimation.
> > > > >
> > > > > As the reviewer mentioned, the absolute magnitude of the underestimation bias in TD3-PIDLag is comparable to the overestimation bias in MICE. However, as discussed in Lines 287-293 of the manuscript, overestimation in the cost value function does not lead to constraint violations compared to underestimation. Moreover, the propagation of overestimation through the cost value function is limited, as the policy tends to avoid actions with high cost estimates. Additionally, overestimation within MICE can effectively correct the estimation bias in high-value regions [1], as shown in the theoretical analysis in Appendix B.1.
> > > > >
> > > > > Lastly, we would like to emphasize that MICE aims to reduce constraint violations caused by the underestimation of the cost value. By learning an intrinsic cost from flashbulb memory,  MICE addresses the underestimation bias and balances policy performance with constraint satisfaction, as demonstrated in Figures 3 and 4.
> > > > >
> > > > >
> > > > >
> > > > > [1] Thommen George Karimpanal, Hung Le, Majid Abdolshah, Santu Rana, Sunil Gupta, Truyen Tran, and Svetha Venkatesh. Balanced q-learning: Combining the influence of optimistic and pessimistic targets. Artificial Intelligence, 325:104021, 2023.

---

> > > > > > ### Author Response · Authors · 2024-12-03
> > > > > >
> > > > > > Dear Reviewer JSW4,
> > > > > >
> > > > > > We kindly hope to know if our response has sufficiently addressed your concerns. As the reviewer’s response period is ending soon, we would greatly appreciate any additional feedback you might have at your earliest convenience. Thank you again for your time and thoughtful review.

---

> ### Comment · Reviewer_JSW4 · 2024-12-03
>
> I thank the reviewer for the explanation. That explains how the authors arrive at the conclusion that overestimation bias of MICE doesn't seem "as bad" as the underestimation bias of TD3.
>
> One thing to highlight is that Fig15 only plots the estimation error throughout training, it doesn't plot the effect it has on safe RL like in Fig3. In fact, the accumulated cost of PID-Lagrangian (purple curve) in CarGoal and PointGoal isn't too bad in Fig3, even though Fig15 shows that PIDLag has significant underestimation. Thus, I'm not entirely sure if the slight underestimation bias of TD3 adversely impacts its ability in adhering to constraint as claimed. Perhaps a similar figure like Fig3 could shed light on this point.

---

> > ### Author Response · Authors · 2024-12-04
> >
> > We greatly appreciate your feedback.
> >
> > Based on your valuable suggestion, we have plotted the training curves for Return and Constraint of TD3-PIDLag, PIDLag, and MICE-PIDLag in the SafetyPointGoal1-v0 and SafetyCarGoal1-v0 environments. The figures are provided in the anonymous link: https://anonymous.4open.science/r/6568-OpenReview/TD3%20vs%20MICE.png.
> >
> > The results show that in SafetyPointGoal1-v0, TD3-PIDLag still presents constraint violations in the later stages of training (from $0.7 * 1e^7$ to $0.9 * 1e^7$ training steps), with the maximum constraint reaching 32, while MICE-PIDLag shows no constraint violations after $0.2 * 1e^7$ training steps.
> >
> > In SafetyCarGoal1-v0, TD3-PIDLag presents constraint violations with a maximum constraint of 28 between $0.3 * 1e^7$ and $0.5 * 1e^7$ training steps, and a maximum constraint of 27 between $0.8 * 1e^7$ and $1.0 * 1e^7$ training steps. In contrast, MICE-PIDLag shows few constraint violations after $0.3 * 1e^7$ training steps.
> > These results demonstrate that the underestimation bias in TD3-PIDLag leads to more constraint violations compared to MICE-PIDLag.
> >
> > Due to the current limitations on uploading revisions, we will incorporate the relevant results in the next revision. We appreciate your time and effort in reviewing our manuscript.

---

### Official Review · Reviewer_MzVx · 2024-11-04

**Soundness:** 3
**Presentation:** 3
**Contribution:** 2
**Rating:** 6
**Confidence:** 4

**Summary:**

This paper presents MICE, a novel constrained reinforcement learning method aimed at alleviating constraint violations. Its main contributions include:

1. Identifying the underestimation of cost value functions as a key factor contributing to constraint violations in constrained reinforcement learning, providing a new perspective.
2. Proposing a memory-based intrinsic cost estimation scheme that enhances the cost estimation of unsafe behaviors by storing unsafe trajectories.
3. Introducing new external-internal cost value functions and optimization objectives, offering theoretical guarantees for convergence and worst-case limits on constraint violations.
4. Experimental results demonstrate that MICE significantly reduces constraint violations while maintaining comparable policy performance to baseline methods, validating its effectiveness and reliability across various environments.

**Strengths:**

1. MICE introduces a novel approach that integrates memory into the CRL framework by drawing an analogy to human cognitive mechanisms. The idea of utilizing a memory module to store and leverage unsafe trajectories offers a fresh perspective for mitigating the underestimation bias of cost value functions.
2. The paper provides a rigorous theoretical foundation for the proposed MICE algorithm, including proofs of convergence and definitions of constraint violation limits, enhancing the robustness of the method.
3. The experimental design is comprehensive, testing across multiple scenarios and demonstrating MICE's effectiveness in reducing constraint violations while maintaining strong performance.
4. The authors emphasize reproducibility by providing code for replicating the results.

**Weaknesses:**

1. Despite experiments across several environments, the evaluation lacks assessments in a broader range of task types and robot types.
2. The proposed method may struggle to scale to high-dimensional inputs, such as predicting costs based on visual input trajectories, as such extensions may incur excessive computational costs.
3. Although this method shows advantages over baselines in balancing constraint satisfaction and performance, it still does not address safety issues during the learning process, merely increasing conservativeness.

**Questions:**

1. The approach for addressing cost underestimation in MICE is heuristic, but Figure 5 indicates that this method can still ensure that actual costs align with thresholds rather than being excessively low. The reviewer is curious about how the authors resolve this issue—specifically, how they maximize costs (implying limits on hazardous behavior) while largely avoiding constraint violations.
2. Could the authors demonstrate the scalability of MICE across more tasks and in more complex settings, particularly with high-dimensional inputs (e.g., trajectories based on images)?

**Details Of Ethics Concerns:**

No ethics concerns

---

> ### Author Response · Authors · 2024-11-23
>
> We sincerely appreciate the reviewer’s insightful comments. Below are our detailed responses to your feedback.
>
> **Weakness 1, 2 and Question 2**
> > Despite experiments across several environments, the evaluation lacks assessments in a broader range of task types and robot types. Could the authors demonstrate the scalability of MICE across more tasks and in more complex settings, particularly with high-dimensional inputs (e.g., trajectories based on images)?
>
> **Response:** Thank you for your valuable suggestion. We have extended our evaluation to encompass more tasks and complex settings, including SafetyPointButton1-v0 and SafetyHopperVelocity-v4, with results presented in Appendix C.2.5 and Figure 16 (highlighted in red for clarity). The SafetyPointButton1-v0 task has more complex settings, with an observation dimensionality of 76 (range $(-\infty, \infty)$) and an action dimensionality of 2 (range $(-\infty, \infty)$). SafetyHopperVelocity-v4 introduces a different robot type.
> These experiments results demonstrate that MICE effectively balances performance and constraint satisfaction compared to multiple baselines, showcasing its scalability across a broader range of task types and robot types.
>
> For high-dimensional state spaces, the random projection layer in the MICE framework compresses trajectories into a latent space, significantly reducing computational complexity. Additionally, the intrinsic generator in MICE further minimizes computational complexity by reducing memory accesses, directly generating the intrinsic cost when the cumulative extrinsic cost of the current trajectory is below the constraint threshold. We only compute the distance between trajectories when the cumulative extrinsic cost of the current trajectory exceeds threshold, thus optimizing the computational costs.
> Thank you for your valuable comments. We are considering scaling MICE to vision-based tasks in future work.
>
>
>
>
> **Weakness 3**
> > Although this method shows advantages over baselines in balancing constraint satisfaction and performance, it still does not address safety issues during the learning process, merely increasing conservativeness.
>
> **Response:** MICE can achieve zero constraint violations during the learning process by increasing the intrinsic factor, as demonstrated in Figures 7(a) and 7(b). Additionally, the extrinsic-intrinsic cost value function in MICE is guaranteed to converge to the optimal value.
>
>
> **Question 1**
> > Figure 5 indicates that this method can still ensure that actual costs align with thresholds rather than being excessively low. How they maximize costs (implying limits on hazardous behavior) while largely avoiding constraint violations?
>
> **Response:** In constrained reinforcement learning (CRL) tasks, the optimization process is complex because maximizing rewards may lead to increased costs, preventing the actual costs from being excessively low. The optimal policy in CRL tasks typically lies near the constraint thresholds, balancing performance and constraint satisfaction.
> The objective of MICE is to maximize cumulative rewards while ensuring cumulative costs remain below the constraint threshold. During maximizing cumulative rewards, if cumulative costs exceed the threshold, indicating constraint violations, MICE reduces the costs to bring them back under the threshold, which ensures the actual costs align with the thresholds while maintaining good reward performance.
>
>
> We hope our responses address your comments. Please let us know if you have follow-up questions. We would be happy to discuss.

---

> > ### Comment · Reviewer_MzVx · 2024-11-25
> >
> > The reviewer thanks the authors for their response, which has addressed my concerns. I am willing to raise my score.

---

> > > ### Author Response · Authors · 2024-11-28
> > >
> > > We sincerely thank you for taking the time to review our response and raising the score to a positive one. We greatly appreciate your consideration.

---

### Author Response · Authors · 2024-11-23
**Global Response**

## Global Response

We sincerely thank all the reviewers for their valuable comments and constructive suggestions.
Guided by these insights, we have conducted additional experiments and analyses. We also made the necessary revisions to the manuscript. For clarity, the revised parts are highlighted in red.

Below, we provide a summary of the modifications made in the revision:

1. The sections on intrinsic cost and flashbulb memory in the main text have been refined to improve clarity and readability.

2. Comparative experiments between the TD3-based cost value function and MICE have been included in Appendix C.2.4.

3. Extended experiments evaluating MICE covering a broader range of task types and robot types have been added to Appendix C.2.5, demonstrating its scalability and adaptability.

4. Experiments with different criteria for judging unsafe trajectories have been incorporated to Appendix C.2.6.

---

### Meta-Review · Area_Chair_M3pS · 2024-12-20

**Metareview:**

This paper presents an interesting approach to constrained reinforcement learning (CRL) by incorporating a memory module. While the idea of MICE is novel and the theoretical foundations are generally strong, the experimental evaluation is somewhat limited in scope, particularly concerning the types of tasks and robot types considered. Furthermore, scalability to high-dimensional inputs like images appears to be a significant concern. The authors also acknowledge that MICE focuses on improving constraint satisfaction without addressing exploration safety, relying on increased conservativeness which can impact learning speed.  Overall, while the core concept holds promise, the limitations in generalizability, scalability, and exploration safety need to be addressed, leading to a borderline reject recommendation for this conference.  Further development and a more comprehensive evaluation are needed before it would be ready for acceptance.

**Additional Comments On Reviewer Discussion:**

Several concerns brought up by the reviewers were addressed during the rebuttal. but the paper will still be benefitted by another revision.

---

### Decision · Program_Chairs · 2025-01-22

Reject